# Federated Submodel Optimization for Hot and Cold Data Features

**Yucheng Ding**[1]     **Chaoyue Niu**[1][*]     **Fan Wu**[1]     **Shaojie Tang**[2]
**Chengfei Lv**[3]     **Yanghe Feng**[4]     **Guihai Chen**[1]

[1]Shanghai Jiao Tong University     [2]University of Texas at Dallas
[3]Alibaba Group     [4]National University of Defense Technology

## Abstract

We focus on federated learning in practical recommender systems and natural language processing scenarios. The global model for federated optimization typically contains a large and sparse embedding layer, while each client's local data tend to interact with part of features, updating only a small submodel with the feature-related embedding vectors. We identify a new and important issue that distinct data features normally involve different numbers of clients, generating the differentiation of hot and cold features. We further reveal that the classical federated averaging algorithm (FedAvg) or its variants, which randomly selects clients to participate and uniformly averages their submodel updates, will be severely slowed down, because different parameters of the global model are optimized at different speeds. More specifically, the model parameters related to hot (resp., cold) features will be updated quickly (resp., slowly). We thus propose federated submodel averaging (FedSubAvg), which introduces the number of feature-related clients as the metric of feature heat to correct the aggregation of submodel updates. We prove that due to the dispersion of feature heat, the global objective is ill-conditioned, and FedSubAvg works as a suitable diagonal preconditioner. We also rigorously analyze FedSubAvg's convergence rate to stationary points. We finally evaluate FedSubAvg over several public and industrial datasets. The evaluation results demonstrate that FedSubAvg significantly outperforms FedAvg and its variants.

## 1  Introduction

Federated learning (FL) [1] allows a large number of clients (e.g., millions of smartphone users) to collaborate in the training of a global machine learning (ML) model under the coordination of a cloud server without sharing raw data. For the production use on the top, Google has deployed FL among the users of its Android keyboard, called Gboard, to polish language models [2]. As the default optimization algorithm of FL at the bottom, federated averaging (FedAvg) averages the participating clients' local new models to update the global model. Much effort of existing work was devoted to proving the convergence of FedAvg, and the key challenge is that the clients' local data are normally non-independent and identically distributed (non-i.i.d.) [3, 4, 5, 6]. One line of work [7, 8, 9, 10, 11] established an $O(1/\sqrt{NT})$ convergence, where $N$ denotes the total number of clients and $T$ denotes the total number of iterations. However, these work required all the clients to participant in each round of FL, which is not practical in cross-device FL. Another line of work [12, 13, 14, 15, 16] allowed partial client participation and proved an $O(1/\sqrt{KT})$ convergence of FedAvg, where $K$ denotes the number of chosen clients in each round. Some other work proposed variants of FedAvg to better deal with data heterogeneity. For example, Li et al. [17] proposed FedProx by adding a

---

[*]Chaoyue Niu (rvince@sjtu.edu.cn) is the corresponding author.

proximal term to local objective; and Karimireddy et al. [18] proposed Scaffold by adding a control variate for each client to control local training.

Besides non-i.i.d. from data distribution, the interactions between the data features and the clients are mutual, partial, and differentiated, especially in practical recommender systems (RS) and natural language processing (NLP) scenarios. First, distinct data features are normally involved by different numbers of clients. For example, the differentiation of popular and unpopular items in RS or hot and cold words in NLP is common. We call such an observation *feature heat dispersion*. Second, a certain client's local data tend to involve a small subspace of the full feature space, which further implies that the client needs to download and update only the feature-related part of the full global model, called a *submodel* in [19]. For example, deep recommendation and language models are normally stacked with a large and sparse embedding layer and some other dense layers, while a client's submodel comprises the full dense layers and the embedding vectors for the client's few local items or words rather than the full and huge embedding layer.

The existing work on FL has not studied the issue of feature heat dispersion yet. However, different parameters of the global model involve distinct features and will be optimized by different numbers of clients and at different speeds, severely deteriorating the performance of FedAvg and its variants. We take an extreme example in RS for illustration. For an unpopular item 1 that appears only in 1% of the clients' (denoted as client group $G_1$) local datasets, the corresponding embedding vector for item 1 is involved in the submodels of those clients in $G_1$. Using conventional FedAvg and its variants, only the clients in $G_1$ will upload non-zero updates, and the update of the embedding vector for item 1 will be slowed down 100 times. In contrast, for a popular item 2 that appears in all the clients' local datasets, the update of the corresponding embedding vector will not be slowed down.

To deal with feature heat dispersion, we propose federated submodel averaging (FedSubAvg), which first averages the local updates of the chosen clients, just like FedAvg, but then multiplies the aggregated update of each model parameter with the ratio between the total number of clients and the number of clients who involve this model parameter. Such a small correction ensures that the expectation of each model parameter's global update is equal to the average of the local updates of the clients who involve this parameter. We theoretically demonstrate the advantage of FedSubAvg over FedAvg and analyze the convergence of FedSubAvg. We first prove that the global objective is ill-conditioned, leading to the slow convergence of FedAvg, and FedSubAvg works as a suitable diagonal preconditioner. We also obtain an $O(\sqrt{N/(n_{\min}KT)})$ convergence with respect to stationary points, where $n_{\min}$ denotes the minimum of the number of clients who involve each individual parameter.

We summarize the key contributions of this work as follows:

- To the best of our knowledge, we are the first to make an in-depth study of FL from the differentiation of hot and cold data features, which is common in practice.

- We identify the defect of FedAvg and its variants in handling feature heat dispersion and propose a novel, effective, and efficient FedSubAvg algorithm.

- We theoretically show that the the global objective is ill-conditioned, and FedSubAvg essentially works as a preconditioner for acceleration. We also give the convergence rate of FedSubAvg in the general non-convex case.

- Using the public MovieLens, Sentiment140, and Amazon datasets, as well as an industrial dataset from Alibaba, we extensively evaluate FedSubAvg[2] and compare it with FedAvg, FedProx, Scaffold, and FedAdam. The evaluation results reveal the superiority of FedSubAvg from faster convergence and smaller train loss.

## 2   Problem Formulation

In this section, we formulate the federated submodel optimization problem with feature heat dispersion in RS and NLP scenarios.

---

[2]The code is available on https://github.com/sjtu-yc/federated-submodel-averaging.

**Optimization Objective.** We consider a distributed optimization setting, in which $N$ clients collaboratively solve the following consensus optimization problem:

$$f\left(\mathbf{X}\right) = \frac{1}{N}\sum_{i=1}^{N}\mathbb{E}_{\xi_i \sim D_i}\left[F\left(\mathbf{X},\xi_i\right)\right] = \frac{1}{N}\sum_{i=1}^{N}f_i\left(\mathbf{X}\right),$$

where $D_i$ denotes client $i$'s local empirical distribution and $\xi_i \sim D_i$ denotes the local training data; $\mathbf{X}$ is the full global model; $F(\mathbf{X},\xi_i)$ is the train loss of $\mathbf{X}$ over the local data $\xi_i$; and $f_i(\mathbf{X})$ is the local empirical error, taking expectation over the randomness of the local data.

**Model Structure and Submodel.** The full recommendation or language model $\mathbf{X}$ normally adopts the network structure of an embedding layer plus some other dense layers, where sparse input features are mapped into embedding vectors, concatenated, and fed into the upper layers. In practice, a client's local data involve only part of the full global model, namely, a submodel, which is related to the client's local data features. For example, in RS (resp., NLP) scenario, client $i$ can use the data collected in the previous week as its local training set $D_i$, and retrieve the submodel in a key-value lookup way, typically, by retrieving a few embedding vectors based on the local item ids (resp., word ids) and directly taking the other dense network layers. Considering the full embedding layer is far beyond any mobile device's capacity, such a submodel design makes FL in these industrial scenarios possible. Formally, we use $S = \{1, 2, \cdots, M\}$ to index the parameters of the full model $\mathbf{X} \in \mathbb{R}^M$ and call it the full index set. We let $S(i) \subseteq S$ denote the index set of client $i$'s submodel $\mathbf{X}_{S(i)}$. In other words, the full model excluding the submodel (i.e., $\mathbf{X}_{S \setminus S(i)}$) does not affect the model output, and the local gradient of $\mathbf{X}_{S \setminus S(i)}$ will always be zero. This further implies that we can rewrite the global objective function in a distributed submodel way:

$$f\left(\mathbf{X}\right) = \frac{1}{N}\sum_{i=1}^{N}f_i\left(\mathbf{X}_{S(i)}\right).$$

The gradient of $f$ is

$$\nabla f\left(\mathbf{X}\right) = \frac{1}{N}\sum_{i=1}^{N}\nabla f_i\left(\mathbf{X}_{S(i)}\right),$$

and the Hessian is

$$\mathbf{H} \triangleq \nabla^2 f\left(\mathbf{X}\right) = \frac{1}{N}\sum_{i=1}^{N}\nabla^2 f_i\left(\mathbf{X}_{S(i)}\right) = \frac{1}{N}\sum_{i=1}^{N}\mathbf{H}_i.$$

Note that when doing any operation (e.g., summation) over multiple submodels, gradients, and Hessians, they will be automatically aligned according to the indices.

**Feature/Parameter Heat Dispersion.** We finally introduce the metric of feature heat dispersion (resp., the resulting parameter heat dispersion), which is defined as the ratio between the maximum and the minimum of the number of clients who involve each individual feature (resp., parameter). Considering the fact that a parameter may involve one or multiple data features in RS and NLP[3], the feature heat dispersion is a lower bound of the parameter heat dispersion. In other words, *high feature heat dispersion inevitably leads to high parameter heat dispersion.* For convenience and clarity in the submodel-level analysis, we mainly take the metric of parameter heat dispersion. We let $n_m$ denote the number of clients involving the parameter with index $m \in S$ and set $n_{\max} = \max_{m \in S} n_m, n_{\min} = \min_{m \in S} n_m$. Then, the parameter heat dispersion is $n_{\max}/n_{\min}$.

## 3 Algorithm Design

In this section, we first show that FedAvg suffers from high parameter heat dispersion in distributed submodel optimization and then propose FedSubAvg to remedy the defect.

---

[3]For example, an embedding vector corresponds to an item in RS or a certain word in NLP, while the dense layers are related to all the items or words.

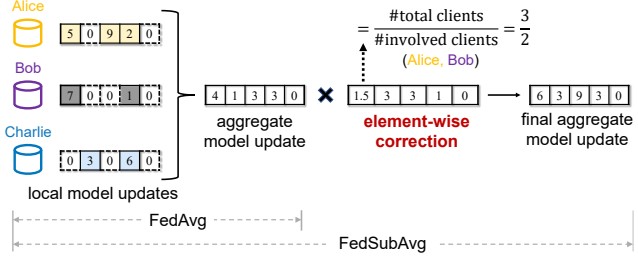

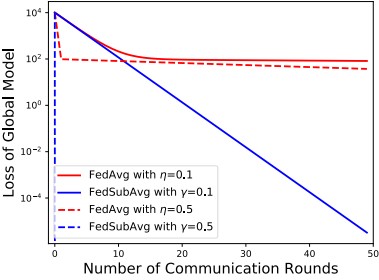

Figure 1: FedAvg vs. FedSubAvg with $N = 3$ clients in total and 2 clients involving the first model parameter. The dashed line indicates that the client does not download the parameter and upload no update.

Figure 2: FedAvg (learning rate $\eta$) vs. FedSubAvg (learning rate $\gamma$) for Example 1 with the parameter heat dispersion of 100.

### 3.1 Slow Convergence of FedAvg

**Example 1.** *We consider a special distributed convex optimization problem with two model parameters, denoted as $w_1$ and $w_2$. Each client $i$'s local data $\xi_i \sim D_i$ are with mean $\mathbf{e}_i = \mathbb{E}[\xi_i] = \mathbf{0}$. In addition, $w_1$ involves only client 1, while $w_2$ involves all the $N$ clients. Then, the parameter heat dispersion is $n_2/n_1 = N$. We formulate this learning problem as minimizing the mean square error:*

$$f((w_1, w_2)) = \frac{1}{N} \sum_{i=1}^{N} f_i \left( (w_1, w_2)_{S(i)} \right),$$

*where $f_1 \left( (w_1, w_2)_{S(1)} \right) = \mathbb{E}_{\xi_1 \sim D_1} \left[ \| (w_1, w_2) - \xi_1 \|^2 \right] = w_1^2 + w_2^2 + \mathbb{E}_{\xi_1 \sim D_1}[\| \xi_1 - \mathbf{e}_1 \|^2]$; and for $i = 2, 3, \cdots, N$, $f_i \left( (w_1, w_2)_{S(i)} \right) = \mathbb{E}_{\xi_i \sim D_i}[(w_2 - \xi_i)^2] = w_2^2 + \mathbb{E}_{\xi_i \sim D_i}[(\xi_i - \mathbf{e}_i)^2]$.*

For this example, the optimal model is $(w_1^*, w_2^*) = (0, 0)$. We leverage FedAvg with only one local iteration and let each client compute the exact (not stochastic) gradient. The learning rate is denoted as $\eta$, and the model is initialized as $(w_1^0, w_2^0)$. After $r$ rounds, the model will become

$$(w_1^r, w_2^r)^\top = \begin{bmatrix} 1 - \frac{2\eta}{N} & 0 \\ 0 & 1 - 2\eta \end{bmatrix}^r (w_1^0, w_2^0)^\top.$$

By choosing $\eta = 0.5$, $(w_1^r, w_2^r) = ((1 - 1/N)^r w_1^0, 0)$, we can find that in the FL scenario with high parameter heat dispersion $N$, $w_1$ will converge at a quite low speed. We also depict the optimization process of FedAvg in Figure 2.

### 3.2 Federated Submodel Averaging

To mitigate parameter heat dispersion, we propose FedSubAvg. As shown in Figure 1, compared with FedAvg, the key principle of FedSubAvg is to further multiply the aggregated update of each model parameter with the ratio between the total number of clients and the number of clients who involve this model parameter[4] (i.e., for the model parameter with index $m$, the correction coefficient is $N/n_m$). We still examine Example 1 for illustration. FedSubAvg will multiply the aggregated update of $w_1$ and $w_2$ with $N$ and 1, respectively. By correction, the model at the $r$-th round is:

$$(w_1^r, w_2^r)^\top = \begin{bmatrix} 1 - 2\gamma & 0 \\ 0 & 1 - 2\gamma \end{bmatrix}^r (w_1^0, w_2^0)^\top,$$

where $\gamma$ is the learning rate of FedSubAvg. Therefore, FedSubAvg converges quickly to the optimal model $(0, 0)$. We also depict the optimization processes of FedSubAvg and FedAvg for Example 1 in Figure 2, when the parameter heat dispersion is 100. We can observe that FedSubAvg greatly outperforms FedAvg from convergence speed and loss.

---

[4]With some privacy preserving methods, we can obtain $n_m$ without revealing the real index set of any client's submodel. Please refer to Appendix F for details.

---

**Algorithm 1** Federated Submodel Averaging (FedSubAvg)

---

**Require:** The total number of clients $N$, the number of local iterations $I$, the number of clients involving the $m$-th model parameter $n_m$, the number of selected clients $K$ per round.

  /* Cloud server's process */

 1: Initializes the global model $\mathbf{X}^1$;

 2: **for** each communication round $r = 1, 2, \cdots, R$ **do**

 3:  Randomly selects $K$ clients, denoted as $C_r$;

 4:  **for** each client $i \in C_r$ **do**

 5:   Receives the index set $S(i)$, returns the submodel $\mathbf{X}^r_{S(i)}$, and receives the submodel update $\Delta \mathbf{x}^r_i$;

 6:  **end for**

 7:  $\mathbf{X}^{r+1} \leftarrow \mathbf{X}^r$;

 8:  **for** each index $m \in \bigcup_{i \in C_r} S(i)$ **do**

 9:   $\mathbf{X}^{r+1}_{\{m\}} \leftarrow \mathbf{X}^r_{\{m\}} + \frac{N}{n_m K} \sum_{i \in C_r} \Delta \mathbf{x}^r_{i,\{m\}}$;

10:  **end for**

11: **end for**

  /* Client $i$'s process */

12: Determines its index set $S(i)$ based on the local data;

13: Uses $S(i)$ to download the submodel $\mathbf{X}^r_{S(i)}$ from the cloud server;

14: Initializes the local submodel $\mathbf{x}^{r,1}_i \leftarrow \mathbf{X}^r_{S(i)}$

15: **for** local iteration $j = 1, 2, \cdots, I$ **do**

16:  $\mathbf{x}^{r,j+1}_i \leftarrow \mathbf{x}^{r,j}_i - \gamma \nabla F(\mathbf{x}^{r,j}_i, \xi_i)$ for $\xi_i \sim D_i$;

17: **end for**

18: Uploads $\Delta \mathbf{x}^r_i = \mathbf{x}^{r,I+1}_i - \mathbf{x}^{r,1}_i$ to the cloud server.

---

We now present the design details of FedSubAvg in Algorithm 1. In each round $r$ of FL, the cloud server first selects $K$ clients to participate (Line 3), denoted as $C_r$. Each selected client $i \in C_r$ determines its index set $S(i)$ of submodel based on the local training set (Line 12). Then, client $i$ uses $S(i)$ to download the submodel $\mathbf{X}^r_{S(i)}$ from the cloud server and initializes the local submodel $\mathbf{x}^{r,1}_i$ (Lines 13–14). Client $i$ locally trains its submodel by doing $I$ iterations of stochastic gradient descent (SGD) (Lines 15–17) and uploads the submodel update $\Delta \mathbf{x}^r_{S(i)}$ (Line 18). After receiving the submodel updates from the selected clients, the cloud server performs aggregation for each index in the union of the participating clients' index sets and updates the global full model (Lines 7–10). In particular, for the global model parameter with index $m$, namely, $\mathbf{X}^r_{\{m\}}$, its expected update, after being corrected with the coefficient $N/n_m$, is

$$\mathbb{E}_{C_r}\left[\Delta \mathbf{X}^r_{\{m\}}\right] = \frac{N}{n_m}\left[\frac{1}{N}\sum_{i=1}^N \Delta \mathbf{x}^r_{i,\{m\}}\right] = \frac{1}{n_m}\sum_{\{i|m\in S(i)\}} \Delta \mathbf{x}^r_{i,\{m\}},$$

which is equal to the average of the local updates of the clients involving it, as required.

## 4 Theoretical Analysis

In this section, we first prove that in FL with high parameter heat dispersion, the global objective $f$ is ill-conditioned. We then prove that optimizing $f$ with FedSubAvg approximates to optimizing a preconditioning objective $\hat{f}$ with gradient descent (GD), thereby remedying ill conditioning. In particular, the diagonal preconditioning matrix comprises of the correction coefficients $\{N/n_m | m \in S\}$ in FedSubAvg and can be obtained without needing to access any client's local raw data and without any expensive computing, which keeps the tenet of FL. Further by analyzing the Hessian of $\hat{f}$, we demonstrate that the superiority of optimizing $\hat{f}$ over $f$. We also obtain the convergence guarantee for FedSubAvg.

### 4.1 FedAvg with Ill-Conditioned Global Objective

We first make an assumption about the eigenvalues of each client's local Hessian.

**Assumption 1** (Bounded Hessian). *For any model* $\mathbf{X}$*, the global Hessian is non-singular, and for each client* $i$*, the Hessian of* $f_i$ *satisfies:*

$$-\rho_2 I_i \preceq \nabla^2 f_i(\mathbf{X}_{S(i)}) \preceq -\rho_1 I_i \ \ or \ \ \rho_1 I_i \preceq \nabla^2 f_i(\mathbf{X}_{S(i)}) \preceq \rho_2 I_i, \ \ with \ \ 0 < \rho_1 < \rho_2.$$

*Further, if* $\mathbf{X}$ *is in a locally convex area of* $f$*, for each parameter* $m$*, more than* $1 - \alpha$ *of the related clients satisfies* $\rho_1 I_i \preceq \nabla^2 f_i(\mathbf{X}_{S(i)}) \preceq \rho_2 I_i$*, where* $\alpha$ *is a constant with* $(1 - \alpha)\rho_1 - \alpha\rho_2 > 0$.

Assumption 1 bounds the eigenvalues of $\nabla^2 f_i(\mathbf{X}_{S(i)})$ and ensures that when $\mathbf{X}$ is in a locally convex area of $f$, $\mathbf{X}_{S(i)}$ is also in a locally convex area of $f_i$ for most of the clients, which helps to obtain the bounds of the global Hessian. Under Assumption 1, we can analyze the condition number of $\mathbf{H}$: $\kappa(\mathbf{H}) \overset{\triangle}{=} \sigma_{\max}(\mathbf{H})/\sigma_{\min}(\mathbf{H})$, where $\sigma_{\max}(\mathbf{H})$ and $\sigma_{\min}(\mathbf{H})$ denote the biggest and smallest singular values of $\mathbf{H}$, respectively.

**Theorem 1.** *Under Assumption 1, for any model* $\mathbf{X}$ *in a locally convex area, the condition number of the global Hessian of* $f$ *satisfies:*

$$\kappa(\mathbf{H}) \geq \frac{n_{\max}\left(\rho_1 - \alpha(\rho_1 + \rho_2)\right)}{n_{\min}\rho_2}.$$

*Proof.* Please refer to Appendix A. $\qquad\qquad\qquad\qquad\qquad\qquad\qquad\qquad\qquad\qquad\qquad\qquad\qquad\square$

Theorem 1 reveals that when the parameter heat dispersion $n_{\max}/n_{\min}$ is high, then the global objective is ill-conditioned. Specifically, in the large-scale RS and NLP scenarios, since the fully connected layer involves all the $N$ clients, while the embedding vector of a cold item or word normally involves a few clients, we have $\kappa(\mathbf{H}) \geq \Theta(N)$. This further implies with a large $N$ in practice, the global objective is extremely ill-conditioned. Therefore, the conventional FedAvg and its variants, which are the approximations to GD, will converge at a quite slow speed, even when the global model is in a locally convex area.

## 4.2 FedSubAvg as a Preconditioner Better than FedAvg

For brevity, we introduce $t$ to denote the global iteration index, thus replacing the round index $r$ and the local iteration index $j$, where $t = (r - 1) \times I + j$. We consider a single iteration of FedSubAvg:

$$\mathbf{X^{t+1}} = \mathbf{X^t} - \gamma \mathbf{D}\left(\frac{1}{N}\sum_{i=1}^{N}\nabla f_i(\mathbf{x}_i^t)\right) \approx \mathbf{X^t} - \gamma \mathbf{D}\nabla f(\mathbf{X^t}), \tag{1}$$

where $\mathbf{X}^t \overset{\triangle}{=} \mathbf{D}(\frac{1}{N}\sum_{i=1}^{N}\mathbf{x}_i^t)$ is defined as the global model at iteration $t$, and $\mathbf{D} = \mathrm{diag}\{N/n_1, N/n_2, \cdots, N/n_M\}$. By treating $\mathbf{D}$ as a preconditioning matrix [20], we can construct a new objective $\hat{f}(\hat{\mathbf{X}}) \overset{\triangle}{=} f(\mathbf{D}^{\frac{1}{2}}\hat{\mathbf{X}}) = f(\mathbf{X})$ with variable $\hat{\mathbf{X}} = \mathbf{D}^{-\frac{1}{2}}\mathbf{X}$. The gradient and the Hessian[5] of $\hat{f}$ are:

$$\nabla\hat{f}(\hat{\mathbf{X}}) = \mathbf{D}^{\frac{1}{2}}\nabla f(\mathbf{X}), \ \ \nabla^2\hat{f}(\hat{\mathbf{X}}) = \mathbf{D}^{\frac{1}{2}}\mathbf{H}\mathbf{D}^{\frac{1}{2}}. \tag{2}$$

Dauphin et al. [20] showed that one SGD update for $\hat{f}$ corresponds to equation 1. Therefore, we can compare FedSubAvg and FedAvg by comparing the characteristics of $\hat{f}$ and $f$.

**Theorem 2.** *Under Assumption 1, for any* $\mathbf{X}$*, the upper bound of the condition number of the corresponding Hessian,* $\hat{\mathbf{H}} \overset{\triangle}{=} \nabla^2\hat{f}(\hat{\mathbf{X}})$*, is always smaller than that of* $\mathbf{H}$*:*

$$\kappa(\mathbf{H}) \leq \frac{\rho_2 n_{\max}}{N\sigma_{\min}(\mathbf{H})} \overset{\triangle}{=} k, \ \ \kappa(\hat{\mathbf{H}}) \leq \frac{\rho_2}{\sigma_{\min}(\hat{\mathbf{H}})} \overset{\triangle}{=} \hat{k} \ \ with \ \ \hat{k} \leq k.$$

*Further, if* $\mathbf{X}$ *in a locally convex area, the condition number of* $\nabla^2\hat{f}(\hat{\mathbf{X}})$*, satisfies:*

$$\kappa(\hat{\mathbf{H}}) \leq \frac{\rho_2}{(\rho_1 - \alpha(\rho_1 + \rho_2))}, \ \ with \ \ \hat{\mathbf{H}} = \nabla^2\hat{f}(\hat{\mathbf{X}}).$$

---

[5]The gradient and the Hessian of $\hat{f}$ are with respect to $\hat{\mathbf{X}}$.

*Proof.* Please refer to Appendix B. □

Theorem 2 indicates that $\hat{\mathbf{H}}$ always has a smaller condition number upper bound compared with $\mathbf{H}$. Further, in a locally convex area, the proposed FedSubAvg significantly reduces the condition number: $\hat{\mathbf{H}}$ is well-conditioned with $\kappa(\mathbf{H}) \leq \Theta(1)$. Therefore, the proposed FedSubAvg, which is an approximation to GD for objective $\hat{f}$, will maintain the efficiency of FedAvg in any case and converge much faster than FedAvg in a locally convex area of $f$.

### 4.3 Convergence Guarantee for FedSubAvg

In this section, we analyze the convergence rate of FedSubAvg in the general non-convex case based on some standard assumptions. We use $\| \nabla \hat{f}(\hat{\mathbf{X}}) \|^2 = \nabla f(\mathbf{X})^\top \mathbf{D} \nabla f(\mathbf{X})$ rather than $\| \nabla f(\mathbf{X}) \|^2$ to characterize the convergence rate. This is because the curvature of the original global objective $f$ has been severely "diluted". In particular, most model parameters involve only a small number of clients, while the zero gradients contributed by many non-involved clients are inaccurately counted in when computing the global gradient[6].

We next make the following assumptions about the objective functions, as well as the variance and the feasible space of the stochastic gradients.

**Assumption 2** (Smoothness). *$f_i(\cdot)$ is L-smooth if*

$$\forall \mathbf{x}, \mathbf{y}, f_i(\mathbf{y}) \leq f_i(\mathbf{x}) + \langle \mathbf{y} - \mathbf{x}, \nabla f_i(\mathbf{x}) \rangle + \frac{L}{2} \| \mathbf{x} - \mathbf{y} \|^2 .$$

**Assumption 3** (Bounded Variance). *During local training, the variance of stochastic gradients on each client is bounded by $\sigma^2$: $\forall i, t : \mathbb{E}_{\xi_i \sim D_i} \left[ \| \nabla f_i(\mathbf{x}_i^t) - \nabla F(\mathbf{x}_i^t, \xi_i) \|^2 \right] \leq \sigma^2$.*

**Assumption 4** (Bounded Gradient Norm). *During local training, the expected $l_2$-norm of the stochastic gradients is bounded by a constant $G^2$: $\forall i, t : \mathbb{E}_{\xi_i \sim D_i} \left[ \| \nabla F(\mathbf{x}_i^t, \xi_i) \|^2 \right] \leq G^2$.*

Assumption 2 is standard. Assumptions 3 and 4 were widely made in the literature [8, 9, 11, 21, 13]. Under these assumptions, we can bound the gradient norm.

**Theorem 3.** *Under Assumptions 2, 3, and 4, we can bound the expected average of the squared gradient norm*

$$\mathbb{E}\left[ \frac{1}{T} \sum_{i=1}^{T} \nabla f(\mathbf{X}^t)^\top \mathbf{D} \nabla f(\mathbf{X}^t) \right] \leq \frac{2 \left( f(\mathbf{X}^1) - f(\mathbf{X}^*) \right)}{\gamma T} + \frac{2\gamma L \sigma^2}{n_{\min}} + \frac{4N\gamma^2 I^2 G^2 L^2}{n_{\min}} + \frac{2\gamma N I^2 G^2 L}{n_{\min} K}.$$

*When $T > n_{\min} K^3 / N$ and $\gamma = \Theta\left( \sqrt{\frac{n_{\min} K}{NT}} \right)$, we have*

$$\mathbb{E}\left[ \frac{1}{T} \sum_{i=1}^{T} \nabla f(\mathbf{X}^t)^\top \mathbf{D} \nabla f(\mathbf{X}^t) \right] \leq O\left( \sqrt{\frac{N}{n_{\min} KT}} \right)$$

*Proof.* Please refer to Appendix C. □

## 5 Evaluation

In this section, we extensively evaluate the performance of FedSubAvg over several datasets with different feature heat dispersion.

### 5.1 Experimental Setups

We choose the following tasks and models for evaluation. The statistics about clients and samples, as well as the feature heat dispersion are shown in Table 1.

**LR for Rating Classification.** We perform a rating classification task over the MovieLens-1M dataset [22], which contains 6,040 clients, 3,883 movies, and 1,000,209 samples. We preprocess the

---

[6]Please refer to Appendix C for detailed explanation.

Table 1: Statistics of four datasets.

|  | # Clients | # Samples | # Samples Per Client | Feature Heat Dispersion |
|---|---|---|---|---|
| MovieLens | 6,040 | 1,000,209 | 165 | 4,331 |
| Sent140 | 1,473 | 79,050 | 54 | 1,451 |
| Amazon | 1,870 | 123,147 | 66 | 232 |
| Alibaba | 49,023 | 16,864,641 | 344 | 3,142 |

dataset to be suitable for binary classification. In particular, the original user ratings of movies range from 0 to 5. We label the samples with the ratings of 4 and 5 to be positive and label the rest to be negative. We randomly select 20% of the samples as the test dataset and leave the remaining 80% as the training dataset for FL. The task is to predict whether users will rate a given movie to be positive based on the user's gender and age and on the movie ID. We first encode gender, age, movie, gender cross movie, and age cross movie, based on the one-hot encoder. Next, the features are input into a logistic regression (LR) model to predict the label.

**LSTM for Sentiment Analysis.** We perform a text sentiment classification task on the Sentiment140 dataset [23], which comprises 1,600,000 tweets collected from 659,775 twitter users. In this task, we use a two-layer long short-term memory (LSTM) network with 100 hidden units and an embedding layer as the binary classifier, where the embedding dimension is set to 25. We naturally partition this dataset by letting each Twitter account correspond to a client. We keep only the clients who hold more than 40 samples and get 1,473 clients in total. We randomly select 20% of the samples as the test dataset and leave the remaining 80% as the training set for FL.

**DIN for CTR Prediction.** We perform a click-through rate (CTR) prediction task on the Amazon electronics dataset and an Alibaba industrial dataset. The Amazon dataset contains 1,689,188 reviews contributed by 192,403 users for 63,001 items. The ratings range from 0 to 5. We label the samples with the rating of 5 to be positive and label the rest to be negative. We naturally partition this dataset by letting each Amazon user correspond to a client. For each client, we take the user ID, the historical sequence of positively rated product as the input to predict the label. We keep only the clients who hold more than 40 samples and get 1,870 clients in total. We select samples with the timestamps more than 1,385,000,000 as the test dataset and leave the remaining samples as the training set. The Alibaba dataset is built from 30-day impression and click logs of 49,023 Taobao clients from June 15, 2019 to July 15, 2019. For a certain Taobao user, we leverage its click behaviors in previous 14 days as historical data to predict its click and non-click behaviors in the following 1 day. We leave out the behaviors within the last 1 day as the target items of the test set while putting the other samples into the training set. For both datasets, we take the deployed deep interest network (DIN) [24] in Alibaba as the model, where the embedding dimension is set to 18.

We use the following five baselines for comparison.

- **FedAvg** averages the local model updates from the participating clients to update the global model.

- **FedProx** is the first variant of FedAvg. The main difference from FedAvg is that FedProx adds a quadratic proximal term to explicitly limit the local model updates. We set the coefficient of the proximal term to 0.01.

- **Scaffold** is another important variant of FedAvg. The key difference from FedAvg is that each client keeps a variate to control the local model updates in Scaffold. However, the size of the control variate is equal to size of the full model, which is prohibitively inefficient for the learning tasks with large-scale full models. Therefore, for the CTR prediction tasks, we make an approximation to Scaffold. In particular, the cloud server performs the controlled update step every round by weighted averaging the historical updates. Please refer to Appendix D.2 for details.

- **FedAdam** is also an important FL algorithm, which adopts an adaptive optimizer [25]. We make more comparisons and discussions in Appendix E.

- **CentralSGD** runs the standard SGD algorithm to train the global model using the whole dataset, sets the number of iterations in each round as $I$, and sets the batch size to the sum of the selected clients' local batch sizes in each round. This ensures the same amount of data per round with the distributed algorithms.

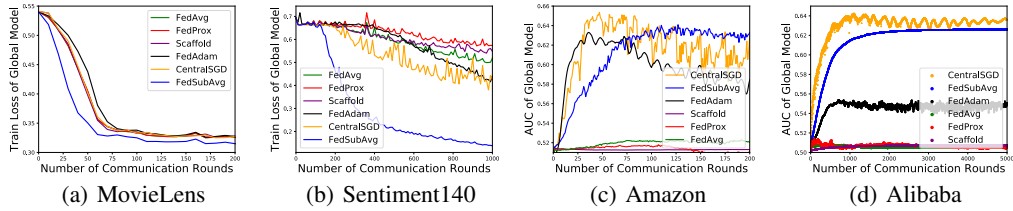

|  | (a) MovieLens | (b) Sentiment140 | (c) Amazon | (d) Alibaba |

Figure 3: Train losses or test AUCs of FedSubAvg and the baselines on different datasets.

Table 2: Number of communication rounds to reach the target train loss or test AUC with different algorithms. #+ indicates that the target was not reached even after # rounds.

|  | CentralSGD | FedAvg | FedProx | Scaffold | FedAdam | FedSubAvg |
|---|---|---|---|---|---|---|
| MovieLens | 180 | 170 | 170 | 180 | 170 | **100** |
| Sent140 | 980 | 1,000+ | 1,000+ | 1,000+ | 1000+ | **260** |
| Amazon | 20 | 200+ | 200+ | 200+ | **16** | 53 |
| Alibaba | **265** | 5,000+ | 5,000+ | 5,000+ | 5000+ | **610** |

Regarding the experimental settings, we choose mini-batch SGD as the optimization algorithm. For the tasks of rating classification and sentiment analysis, $K = 50$ clients are randomly chosen per round as default; and for the CTR prediction tasks, $K$ is set to 100 as default. The settings of the other hyperparameters are deferred to Appendix D.3.

## 5.2 Evaluation Results

We first present the results of FedSubAvg and the baselines under the default $K$. We then vary $K$ to show its impact.

**FedSubAvg vs. Baselines.** For the rating classification on the MovieLens dataset and the sentiment analysis on the Sent140 dataset, we plot the train loss in Figure 3(a) and Figure 3(b); and for the CTR prediction on Amazon and the Alibaba dataset, we plot the test area under the curve (AUC)[7] in Figure 3(c) and 3(d). In addition, we measure the convergence rates of different algorithms by counting the communication rounds to reach a target train loss or test AUC. We set the target loss in the rating classification (resp., the sentiment analysis) to be the minimum loss of CentralSGD, which is 0.325 (resp., 0.380); and we set the target test AUC to be 0.6 in two CTR prediction tasks. The results are listed in Table 2.

From Figure 3 and Table 2, we observe that FedSubAvg consistently outperforms FedAvg and its variants. Specifically, (1) in the rating classification, FedSubAvg always has the smallest train loss during FL and reaches the target at the 100-th communication round, $1.7\times$ faster than FedAvg, FedProx, and FedAdam, and $1.8\times$ faster than Scaffold and CentralSGD; (2) in the sentiment analysis, FedSubAvg still has the smallest train loss during FL and reaches the target at the 260-th round, $3.77\times$ faster than CentralSGD, while FedAvg, FedProx, Scaffold, and FedAdam cannot reach the target even in 1,000 rounds; (3) in the CTR prediction on the Amazon dataset, FedSubAvg achieves the highest AUC while FedAdam achieves the target AUC first among all the FL algorithms. FedSubAvg achieves the highest AUC of 0.641 in 200 rounds, decreasing by 0.013 in terms of AUC compared with the ideal CentralSGD. In contrast, FedAvg achieves the highest AUC of 0.523, FedProx achieves the highest AUC of 0.519, Scaffold achieves the highest AUC of 0.514, and FedAdam achieves the highest AUC of 0.633. In addition, FedSubAvg reaches the target test AUC at the 53-th round, while FedAvg, FedProx, and Scaffold cannot reach the target even after 200 rounds; and (4) in the CTR prediction on the Alibaba dataset, FedSubAvg still outperforms all the other FL algorithms. FedSubAvg achieves the highest AUC of 0.626 in 5,000 rounds, decreasing by 0.016 compared with

---

[7]The positive and negative samples in CTR datasets (especially the Alibaba dataset) are extremely uneven. Even if all the samples are predicted to be negative (or positive), the train loss is very small. As a result, the train losses of different algorithms are hard to distinguish, and we choose to plot the test AUCs instead.

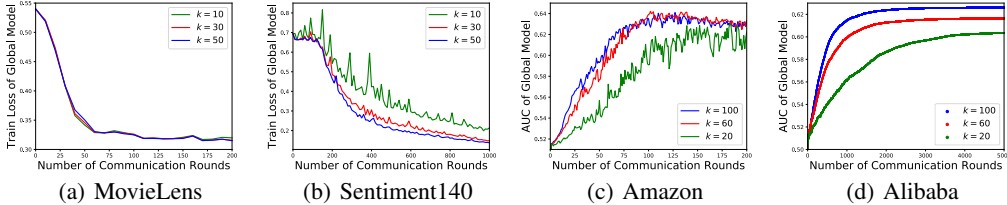

| | (a) MovieLens | (b) Sentiment140 | (c) Amazon | (d) Alibaba |

Figure 4: Train losses or test AUCs of FedSubAvg on different datasets with the varying number of selected clients per round $K$.

Table 3: Number of communication rounds for FedSubAvg to reach the target train loss or test AUC with the varying number of selected clients per round $K$.

| | MovieLens | | | Sent140 | | | Amazon | | | Alibaba | | |
|---|---|---|---|---|---|---|---|---|---|---|---|---|
| $K$ | 10 | 30 | 50 | 10 | 30 | 50 | 20 | 60 | 100 | 20 | 60 | 100 |
| Rounds | 100 | 110 | 100 | 420 | 270 | 260 | 95 | 57 | 53 | 3,469 | 1,030 | 610 |

CentralSGD. In contrast, FedProx achieves the highest AUC of 0.514, FedAvg achieves the highest AUC of 0.509, Scaffold achieves the highest AUC of 0.507, and FedAdam achieves the highest AUC of 0.554. Moreover, FedSubAvg reaches the target test AUC at the 610-th round, whereas the other FL algorithms cannot reach the target even after 5,000 rounds.

**Impact of participating clients.** We next evaluate the impact of the number of selected clients $K$ per round on FedSubAvg. We set $K = 10$, 30, and 50 for the rating classification and the sentiment analysis, and set $K = 20$, 60, and 100 for the CTR prediction. We plot the results in Figure 4 and record the minimum number of rounds to reach the target in Table 3. We observe that FedSubAvg with a larger $K$ generally converges much faster, which validates the speedup with respect to $K$. (1) In the rating classification, FedSubAvg with different $K$ behaves somewhat uniformly. This is because a larger $K$ improves the convergence by reducing the variance of the global model update, while for such a simple convex optimization scenario, the variance is already small enough with $K = 10$; (2) in the sentiment analysis, FedSubAvg with $K = 50$ reaches the target train loss $1.62\times$ faster than FedSubAvg with $K = 10$; (3) in the CTR prediction on the Amazon dataset , FedSubAvg with $K = 100$ reaches the target test AUC $1.79\times$ faster than FedSubAvg with $K = 20$; and (4) in the CTR prediction on the Alibaba dataset, FedSubAvg with $K = 100$ reaches the target test AUC $5.69\times$ faster than FedSubAvg with $K = 20$.

## 6 Conclusion

In this work, we studied federated submodel optimization over non-i.i.d. data with feature heat dispersion. We proposed FedSubAvg, which ensures the expectation of the global update of each model parameter to be equal to the average of the local updates of the clients who involve it. We also proved that FedSubAvg works as a preconditioner to improve collaborative optimization and thoroughly analyzed the convergence. Empirical studies demonstrated the remarkable superiority of FedSubAvg over FedAvg and its variants.

## Acknowledgment

This work was supported in part by National Key R&D Program of China No. 2019YFB2102200, in part by China NSF grant No. 62202296, 62025204, 62072303, 61972252, 61972254, and 61832005, in part by Alibaba Group through Alibaba Innovation Research (AIR) Program, and in part by Tencent Rhino Bird Key Research Project. The opinions, findings, conclusions, and recommendations expressed in this paper are those of the authors and do not necessarily reflect the views of the funding agencies or the government.

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
