# A Proof of Theorem 1

## A.1 Proof Sketch

We first introduce the following lemma:

**Lemma 1.** *For matrices* $\mathbf{A_1}, \mathbf{A_2}, \mathbf{B_1}, \mathbf{B_2} \in \mathbb{M}_n$, *if* $\mathbf{A_1} \preceq \mathbf{B_1}$ *and* $\mathbf{A_2} \preceq \mathbf{B_2}$, *then we have* $\mathbf{A_1} + \mathbf{A_2} \preceq \mathbf{B_1} + \mathbf{B_2}$.

By Lemma 1, we have

$$
\begin{aligned}
\mathbf{H} = \frac{1}{N}\sum_{i=1}^N \mathbf{H}_i &\succeq (\rho_1 - \alpha(\rho_1 + \rho_2)) \cdot \frac{1}{N}\sum_{i=1}^N I_i \\
&\overset{(a)}{=} \frac{(\rho_1 - \alpha(\rho_1 + \rho_2))}{N}
\begin{bmatrix}
n_1 & 0 & \cdots & 0 \\
0 & n_2 & \cdots & 0 \\
\cdots & \cdots & \cdots & \cdots \\
0 & 0 & \cdots & n_M
\end{bmatrix}
\overset{\triangle}{=} \mathbf{M}_1,
\end{aligned}
\tag{3}
$$

where (a) follows from the align operation when doing summation over local Hessians.

Similarly, we have

$$
\mathbf{H} = \frac{1}{N}\sum_{i=1}^N \mathbf{H}_i \preceq \frac{1}{N}\sum_{i=1}^N \rho_2 I_i = \frac{\rho_2}{N}
\begin{bmatrix}
n_1 & 0 & \cdots & 0 \\
0 & n_2 & \cdots & 0 \\
\cdots & \cdots & \cdots & \cdots \\
0 & 0 & \cdots & n_M
\end{bmatrix}
\overset{\triangle}{=} \mathbf{M}_2.
\tag{4}
$$

Thus, $\mathbf{H} \preceq \mathbf{M}_2$. We next introduce Lemma 2 about eigenvalue.

**Lemma 2.** *For matrices* $\mathbf{A}, \mathbf{B} \in \mathbb{M}_n$, *if* $\mathbf{A} \preceq \mathbf{B}$, *then we have* $\lambda_{\min}(\mathbf{A}) \le \lambda_{\min}(\mathbf{B})$ *and* $\lambda_{\max}(\mathbf{A}) \le \lambda_{\max}(\mathbf{B})$, *where* $\lambda_{\max}(\cdot)$ *(resp.,* $\lambda_{\min}(\cdot)$*) denotes taking the maximum (resp., minimum) eigenvalue..*

By Lemma 2, we have $\lambda_{\max}(\mathbf{H}) \ge \lambda_{\max}(\mathbf{M}_1) = n_{\max}(\rho_1 - \alpha(\rho_1 + \rho_2))/N$, and $\lambda_{\min}(\mathbf{H}) \le \lambda_{\min}(\mathbf{M}_2) = n_{\min}\rho_2/N$. Further, for the positive-definite Hessian $\mathbf{H}$, we have $\sigma_{\max}(\mathbf{H}) = \lambda_{\max}(\mathbf{H})$ and $\sigma_{\min}(\mathbf{H}) = \lambda_{\min}(\mathbf{H})$. Therefore, we have the lower bound of the condition number of $\mathbf{H}$:

$$
\kappa(\mathbf{H}) \ge \frac{n_{\max}(\rho_1 - \alpha(\rho_1 + \rho_2))/N}{n_{\min}\rho_2/N} = \frac{n_{\max}(\rho_1 - \alpha(\rho_1 + \rho_2))}{n_{\min}\rho_2} = \Theta\left(\frac{n_{\max}}{n_{\min}}\right).
\tag{5}
$$

## A.2 Proof of Lemmas

*Proof of Lemma 1.* If $\mathbf{A_1} \preceq \mathbf{B_1}$ and $\mathbf{A_2} \preceq \mathbf{B_2}$, for any $x \in \mathbb{R}^n$, we have

$$
x^\top \mathbf{A}_1 x \le x^\top \mathbf{B}_1 x, \quad x^\top \mathbf{A}_2 x \le x^\top \mathbf{B}_2 x.
\tag{6}
$$

Thus, $\forall x \in \mathbb{R}^n$, we have

$$
x^\top (\mathbf{A}_1 + \mathbf{A}_2) x \le x^\top (\mathbf{B}_1 + \mathbf{B}_2) x,
\tag{7}
$$

and we further have $(\mathbf{A}_1 + \mathbf{A}_2) \preceq (\mathbf{B}_1 + \mathbf{B}_2)$. $\qquad\square$

*Proof of Lemma 2.* For any matrix $\mathbf{P} \in \mathbb{M}_n$ with $\mathbf{P}^\top = \mathbf{P}$, we have

$$
\lambda_{\max}(\mathbf{P}) = \max_{x \in \mathbb{R}^n, x \neq \mathbf{0}}\{\frac{x^\top \mathbf{P} x}{x^\top x}\}, \quad \lambda_{\min}(\mathbf{P}) = \min_{x \in \mathbb{R}^n, x \neq \mathbf{0}}\{\frac{x^\top \mathbf{P} x}{x^\top x}\}.
\tag{8}
$$

For two matrices $\mathbf{A}, \mathbf{B}$ with $\mathbf{A} \preceq \mathbf{B}$, we have

$$
\frac{x^\top \mathbf{A} x}{x^\top x} \le \frac{x^\top \mathbf{B} x}{x^\top x},
\tag{9}
$$

for any vector $x \in \mathbb{R}^n (x \neq \mathbf{0})$. Therefore, we have

$$
\max_{x \in \mathbb{R}^n, x \neq \mathbf{0}}\{\frac{x^\top \mathbf{A} x}{x^\top x}\} \le \max_{x \in \mathbb{R}^n, x \neq \mathbf{0}}\{\frac{x^\top \mathbf{B} x}{x^\top x}\}, \quad \min_{x \in \mathbb{R}^n, x \neq \mathbf{0}}\{\frac{x^\top \mathbf{A} x}{x^\top x}\} \le \min_{x \in \mathbb{R}^n, x \neq \mathbf{0}}\{\frac{x^\top \mathbf{B} x}{x^\top x}\}.
\tag{10}
$$

So we have $\lambda_{\max}(\mathbf{A}) \le \lambda_{\max}(\mathbf{B})$ and $\lambda_{\min}(\mathbf{A}) \le \lambda_{\min}(\mathbf{B})$. $\qquad\square$

# B  Proof of Theorem 2

The Hessian of $\hat{f}$ is $\hat{\mathbf{H}} \overset{\triangle}{=} \mathbf{D}^{\frac{1}{2}}\mathbf{H}\mathbf{D}^{\frac{1}{2}}$.

We first consider the condition number of $\hat{\mathbf{H}}$ when $\mathbf{X}$ is in a locally convex area. By equations 3 and 4, we have $\mathbf{M}_1 \preceq \mathbf{H} \preceq \mathbf{M}_2$. Rearranging the terms yields $\mathbf{H} - \mathbf{M}_1 \succeq 0$ and $\mathbf{M}_2 - \mathbf{H} \succeq 0$. Therefore, for any vector $x \in \mathbb{R}^M$, we have

$$
\begin{aligned}
x^\top \left( \hat{\mathbf{H}} - \mathbf{D}^{\frac{1}{2}}\mathbf{M}_1\mathbf{D}^{\frac{1}{2}} \right) x &= x^\top \mathbf{D}^{\frac{1}{2}} \left( \mathbf{H} - \mathbf{M}_1 \right) \mathbf{D}^{\frac{1}{2}} x = \left( \mathbf{D}^{\frac{1}{2}} x \right)^\top \left( \mathbf{H} - \mathbf{M}_1 \right) \left( \mathbf{D}^{\frac{1}{2}} x \right) \geq 0, \\
x^\top \left( \mathbf{D}^{\frac{1}{2}}\mathbf{M}_2\mathbf{D}^{\frac{1}{2}} - \hat{\mathbf{H}} \right) x &= x^\top \mathbf{D}^{\frac{1}{2}} \left( \mathbf{M}_2 - \mathbf{H} \right) \mathbf{D}^{\frac{1}{2}} x = \left( \mathbf{D}^{\frac{1}{2}} x \right)^\top \left( \mathbf{M}_2 - \mathbf{H} \right) \left( \mathbf{D}^{\frac{1}{2}} x \right) \geq 0.
\end{aligned}
\tag{11}
$$

So we have

$$
\mathbf{D}^{\frac{1}{2}}\mathbf{M}_1\mathbf{D}^{\frac{1}{2}} \preceq \hat{\mathbf{H}} \preceq \mathbf{D}^{\frac{1}{2}}\mathbf{M}_2\mathbf{D}^{\frac{1}{2}}.
\tag{12}
$$

By Lemma 2, we have

$$
\begin{aligned}
\lambda_{\min}\left( \hat{\mathbf{H}} \right) &\geq \lambda_{\min}\left( \mathbf{D}^{\frac{1}{2}}\mathbf{M}_1\mathbf{D}^{\frac{1}{2}} \right) = (\rho_1 - \alpha(\rho_1 + \rho_2)), \\
\lambda_{\max}\left( \hat{\mathbf{H}} \right) &\leq \lambda_{\max}\left( \mathbf{D}^{\frac{1}{2}}\mathbf{M}_2\mathbf{D}^{\frac{1}{2}} \right) = \rho_2.
\end{aligned}
\tag{13}
$$

Thus, the condition number of $\hat{\mathbf{H}}$ satisfies $\kappa(\hat{\mathbf{H}}) \leq \rho_2/(\rho_1 - \alpha(\rho_1 + \rho_2)) = \Theta(1)$.

We next consider the minimum singular value of $\mathbf{H}$ and $\hat{\mathbf{H}}$ with $\sigma_{\min}(\mathbf{H}) = \sqrt{\lambda_{\min}(\mathbf{H}^2)}$ and $\sigma_{\min}(\hat{\mathbf{H}}) = \sqrt{\lambda_{\min}(\hat{\mathbf{H}}^2)}$ in any case. Let $x_0 \in \mathbb{R}^M (x_0 \neq \mathbf{0})$ such that $\lambda_{\min}(\hat{\mathbf{H}}^2) = x_0^\top \hat{\mathbf{H}}^2 x_0$. Let $x_1 = \mathbf{D}^{\frac{1}{2}} x_0$. Then, we have

$$
\lambda_{\min}\left( \mathbf{H}^2 \right) \leq \frac{x_1^\top \mathbf{H}^2 x_1}{x_1^\top x_1} = \frac{x_0^\top \mathbf{D}^{\frac{1}{2}}\mathbf{H}^2\mathbf{D}^{\frac{1}{2}} x_0}{x_0^\top \mathbf{D} x_0} \overset{(a)}{\leq} \frac{(n_{\max}/N)\lambda_{\min}(\hat{\mathbf{H}}^2)}{(N/n_{\max})x_0^\top x_0} = \left( \frac{n_{\max}}{N} \right)^2 \lambda_{\min}\left( \hat{\mathbf{H}}^2 \right),
\tag{14}
$$

where (a) follows from $\lambda_{\min}(\hat{\mathbf{H}}^2) = x_0^\top \hat{\mathbf{H}}^2 x_0 = x_0^\top \mathbf{D}^{\frac{1}{2}}\mathbf{H}\mathbf{D}\mathbf{H}\mathbf{D}^{\frac{1}{2}} x_0 \geq \frac{N}{n_{\max}} x_0^\top \mathbf{D}^{\frac{1}{2}}\mathbf{H}^2\mathbf{D}^{\frac{1}{2}} x_0$. Therefore, we have $\sigma_{\min}(\mathbf{H}) \leq \frac{n_{\max}}{N}\sigma_{\min}(\hat{\mathbf{H}})$.

Under Assumption 1 and equation 4, we have $\mathbf{H} \preceq \mathbf{M}_2$. Similarly, we can obtain $\mathbf{H} \succeq -\mathbf{M}_2$. By Lemma 2, we further have

$$
\lambda_{\max}(\mathbf{H}) \leq \lambda_{\max}(\mathbf{M}_2) = \frac{n_{\max}\rho_2}{N}, \quad \lambda_{\min}(\mathbf{H}) \geq \lambda_{\min}(-\mathbf{M}_2) = -\frac{n_{\max}\rho_2}{N}.
\tag{15}
$$

Therefore, we have $\sigma_{\max}(\mathbf{H}) \leq \frac{n_{\max}\rho_2}{N}$. Similar to equation 11, we have

$$
\begin{aligned}
\lambda_{\max}\left( \hat{\mathbf{H}} \right) &= \lambda_{\max}\left( \mathbf{D}^{\frac{1}{2}}\mathbf{H}\mathbf{D}^{\frac{1}{2}} \right) \leq \lambda_{\max}\left( \mathbf{D}^{\frac{1}{2}}\mathbf{M}_2\mathbf{D}^{\frac{1}{2}} \right) = \rho_2, \\
\lambda_{\min}\left( \hat{\mathbf{H}} \right) &= \lambda_{\min}\left( \mathbf{D}^{\frac{1}{2}}\mathbf{H}\mathbf{D}^{\frac{1}{2}} \right) \geq \lambda_{\min}\left( -\mathbf{D}^{\frac{1}{2}}\mathbf{M}_2\mathbf{D}^{\frac{1}{2}} \right) = -\rho_2.
\end{aligned}
\tag{16}
$$

Thus, we have $\sigma_{\max}(\hat{\mathbf{H}}) \leq \rho_2$, and the upper bound of the eigenvalues of $\mathbf{H}$ and $\hat{\mathbf{H}}$ are:

$$
\kappa\left( \mathbf{H} \right) \leq \frac{\rho_2 n_{\max}}{N\sigma_{\min}(\mathbf{H})} \overset{\triangle}{=} k, \quad \kappa\left( \hat{\mathbf{H}} \right) \leq \frac{\rho_2}{\sigma_{\min}(\hat{\mathbf{H}})} \overset{\triangle}{=} \hat{k}.
\tag{17}
$$

With $\sigma_{\min}(\mathbf{H}) \leq \frac{n_{\max}}{N}\sigma_{\min}(\hat{\mathbf{H}})$, we have $\hat{k} \leq k$.

## C  Proof of Theorem 3

**C.1**  $\| \nabla \hat{f}(\hat{\mathbf{X}}) \|^2$ **vs.** $\| \nabla f(\mathbf{X}) \|^2$

In this section, we explain why we use $\| \nabla \hat{f}(\hat{\mathbf{X}}) \|^2$ rather than $\| \nabla f(\mathbf{X}) \|^2$ to characterize the convergence rate. In general, it is hard to develop a convergence rate for objective values. However, when the global model is in a locally convex area of $f$, we can obtain the relationship between the gradient and the local optimum. We first show the relationship between $\| \nabla f(\mathbf{X}) \|^2$ and the local optimum in the scenarios **without parameter heat dispersion** (i.e., each client's local data involve the full global model and $n_m = N, \ \forall m \in S$).

**Theorem 4.** *When there is no parameter heat dispersion, and $\mathbf{X}$ is in a $\mu$-strongly convex area of $f_i$ for each $i$, if $\| \nabla f(\mathbf{X}) \|^2 \leq \epsilon$, we have $f(\mathbf{X}) \leq f(\mathbf{X}^*_{local}) + \frac{\epsilon}{2\mu}$, where $\mathbf{X}^*_{local}$ is the local optimum.*

*Proof.* For any model $\mathbf{Y}$ and $\mathbf{X}$ the area, by the $\mu$ local convexity of $f_i$, we have

$$f_i(\mathbf{Y}) \geq f_i(\mathbf{X}) + \langle \mathbf{Y} - \mathbf{X}, \nabla f_i(\mathbf{X}) \rangle + \frac{\mu}{2} \| \mathbf{Y} - \mathbf{X} \|^2. \tag{18}$$

Summing the right-hand of the inequality over $i = \{1, 2, \cdots, N\}$ and dividing it by $N$ yields

$$T(\mathbf{Y}) \triangleq f(\mathbf{X}) + \langle \mathbf{Y} - \mathbf{X}, \nabla f(\mathbf{X}) \rangle + \frac{\mu}{2} \| \mathbf{Y} - \mathbf{X} \|^2, \tag{19}$$

where $T(\mathbf{Y})$ is a quadratic function of $\mathbf{Y}$, we have

$$T(\mathbf{Y}) \overset{(a)}{\geq} f(\mathbf{X}) - \frac{1}{2\mu} \| \nabla f(\mathbf{X}) \|^2, \tag{20}$$

where (a) equals when $\mathbf{Y} = \mathbf{X} - \frac{1}{\mu} \nabla f(\mathbf{X})$, so we have $f(\mathbf{X}) \leq f(\mathbf{Y}) + \frac{1}{2\mu} \| \nabla f(\mathbf{X}) \|^2$ for any $\mathbf{Y}$. Therefore, if $\| \nabla f(\mathbf{X}) \|^2 \leq \epsilon$, we have $f(\mathbf{X}) \leq f(\mathbf{X}^*_{local}) + \frac{1}{2\mu} \| \nabla f(\mathbf{X}) \|^2 \leq f(\mathbf{X}^*_{local}) + \frac{\epsilon}{2\mu}$. □

We next show the relationship of $\| \nabla f(\mathbf{X}) \|^2$, $\| \nabla \hat{f}(\hat{\mathbf{X}}) \|^2$, and the local optimum when **considering parameter heat dispersion**.

**Theorem 5.** *Under Assumption 1, when $\mathbf{X}$ is in a locally convex area of $f$, if $\| \nabla \hat{f}(\hat{\mathbf{X}}) \|^2 \leq \epsilon$, we have $f(\mathbf{X}) \leq f(\mathbf{X}^*_{local}) + \frac{\epsilon}{2\mu_0}$, where $\mu_0 = \rho_1 - \alpha(\rho_1 + \rho_2) > 0$. However, if $\| \nabla f(\mathbf{X}) \|^2 \leq \epsilon$, we can only guarantee that $f(\mathbf{X}) \leq f(\mathbf{X}^*_{local}) + \frac{N\epsilon}{2n_{\min}\mu_0}$.*

*Proof.* We use $\mathbf{X}_i$ to denote $\mathbf{X}_{S(i)}$ for short. For any static model $\mathbf{Y}$ and $\mathbf{X}$, by the $\mu_i$ local convexity of $f_i$, we have

$$f_i(\mathbf{Y}_i) \geq f_i(\mathbf{X}_i) + \langle \mathbf{Y}_i - \mathbf{X}_i, \nabla f_i(\mathbf{X}_i) \rangle + \frac{\mu_i}{2} \| \mathbf{Y}_i - \mathbf{X}_i \|^2, \tag{21}$$

where $\mu_i$ denotes the minimum eigenvalue of $\mathbf{H}_i$. By Assumption 1, we have $\sum_{m \in S(i)} \mu_i \geq n_m(\rho_1 - \alpha(\rho_1 + \rho_2)) = n_m \mu_0$ for any parameter $m$. Summing the right-hand of the inequality over $i = \{1, 2, \cdots, N\}$ yields

$$T(\mathbf{Y}) \triangleq \sum_{i=1}^{N} \left[ f_i(\mathbf{X}_i) + \langle \mathbf{Y}_i - \mathbf{X}_i, \nabla f_i(\mathbf{X}_i) \rangle + \frac{\mu_i}{2} \| \mathbf{Y}_i - \mathbf{X}_i \|^2 \right], \tag{22}$$

where $T(\mathbf{Y})$ is a quadratic function of $\mathbf{Y}$:

$$
\begin{aligned}
T(\mathbf{Y}) &= \sum_{i=1}^{N} f_i(\mathbf{X}_i) + \sum_{m=1}^{M} \sum_{m \in S(i)} \left[ \frac{\mu_i}{2}(y_m - x_m)^2 + \frac{\partial f_i}{\partial x_m}(y_m - x_m) \right] \\
&\geq \sum_{i=1}^{N} f_i(\mathbf{X}_i) + \sum_{m=1}^{M} \left[ \frac{\mu_0 n_m}{2}(y_m - x_m)^2 + \sum_{m \in S(i)} \frac{\partial f_i}{\partial x_m}(y_m - x_m) \right] \\
&\overset{(a)}{\geq} \sum_{i=1}^{N} f_i(\mathbf{X}_i) - \frac{1}{2\mu_0} \sum_{m=1}^{M} \frac{1}{n_m} \left( \sum_{m \in S(i)} \frac{\partial f_i}{\partial x_m} \right)^2 \\
&= N f(\mathbf{X}) - \frac{N}{2\mu_0} \nabla f(\mathbf{X})^\top \mathbf{D} \nabla f(\mathbf{X}),
\end{aligned}
\tag{23}
$$

where $x_m$ and $y_m$ denote the parameter $m$ of model $\mathbf{X}$ and $\mathbf{Y}$, respectively, and (a) equals when $y_m = x_m - \frac{1}{\mu_0 n_m} \sum_{m \in S(i)} \partial f_i / \partial x_m$. Since $f(\mathbf{X}) = \frac{1}{N} \sum_{i=1}^{N} f_i(\mathbf{X}_i)$, we have

$$f(\mathbf{Y}) \geq f(\mathbf{X}) - \frac{1}{2\mu_0} \parallel \nabla \hat{f}(\hat{\mathbf{X}}) \parallel^2 . \tag{24}$$

If $\parallel \nabla \hat{f}(\hat{\mathbf{X}}) \parallel^2 < \epsilon$, by letting $\mathbf{Y} = \mathbf{X}^*_{local}$, we have $f(\mathbf{X}) \leq f(\mathbf{X}^*) + \frac{\epsilon}{2\mu_0}$.

On the contrary, if $\parallel \nabla f(\mathbf{X}) \parallel^2 < \epsilon$, since

$$f(\mathbf{X}^*_{local}) \geq f(\mathbf{X}) - \frac{1}{2\mu_0} \parallel \nabla \hat{f}(\hat{\mathbf{X}}) \parallel^2 \geq f(\mathbf{X}) - \frac{N}{2n_{\min}\mu_0} \parallel \nabla f(\mathbf{X}) \parallel^2,$$

we can only guarantee that $f(\mathbf{X}) \leq f(\mathbf{X}^*_{local}) + \frac{N\epsilon}{2n_{\min}\mu_0}$. $\qquad\square$

We note that there is a difference between equation 18 and 21: for each client $i$, equation 18 involves all the parameters of the full model while equation 21 involves only partial parameters of the submodel, which causes a change in the lower bound of $T(\mathbf{Y})$ and further leads to a change of conclusion.

By Theorem 5, we can also show the superiority of FedSubAvg over FedAvg. The existing work proved an $O(\sqrt{1/KT})$ convergence of $\parallel \nabla f(\mathbf{X}^T) \parallel^2$ in FedAvg. When $\parallel \nabla f(\mathbf{X}^T) \parallel^2 \leq O(\sqrt{1/KT})$ and $\mathbf{X}^T$ is in a locally convex area of $f$, we have $f(\mathbf{X}^T) \leq f(\mathbf{X}^*_{local}) + O(\frac{N}{n_{\min}\sqrt{KT}})$. In contrast, we proved an $O(\sqrt{\frac{N}{n_{min}KT}})$ convergence of $\parallel \nabla \hat{f}(\hat{\mathbf{X}}) \parallel^2$ in FedSubAvg. When $\parallel \nabla \hat{f}(\hat{\mathbf{X}}^T) \parallel^2 \leq O(\sqrt{1/KT})$ and $\mathbf{X}^T$ is in a locally convex area of $f$, we have $f(\mathbf{X}^T) \leq f(\mathbf{X}^*_{local}) + O(\sqrt{\frac{N}{n_{\min}KT}})$, which indicates that compared to FedAvg, FedSubAvg converges $\Theta(\sqrt{N/n_{\min}})$ times faster for the objective value.

## C.2 Additional Notations

Let $\mathbf{X}_i$ denote $\mathbf{X}_{S(i)}$ and $\mathbf{U}$ denote $\frac{1}{N} \cdot \mathbf{D} = \mathrm{diag}\{1/n_1, 1/n_2, \cdots, 1/n_M\}$. We have

$$\mathbf{X}^t = \mathbf{U} \cdot \sum_{i=1}^{N} \mathbf{x}_i^t \tag{25}$$

We then assume that FedSubAvg always activates all the clients at the beginning of each communication round and then uses the parameters maintained by a few selected clients to generate the next-round parameter. It is clear that this update scheme is equivalent to the original. Then, the update of FedSubAvg can be summarized as: for all $i \in [N]$,

$$\mathbf{y}_i^{t+1} = \mathbf{x}_i^t - \gamma \mathbf{g}_i^t, \tag{26}$$

$$\mathbf{x}_i^{t+1} = \begin{cases} \mathbf{y}_i^{t+1} & \text{if } t \text{ is not a multiple of } I, \\ \mathbf{X}_i^{t+1-I} + \mathbf{U}_i \cdot \frac{N}{K} \sum_{j \in C_{t+1}} \left( \mathbf{y}_j^{t+1} - \mathbf{x}_j^{t+1-I} \right) & \text{if } t \text{ is a multiple of } I, \end{cases} \tag{27}$$

where $\mathbf{g}_i^t \triangleq \nabla F(\mathbf{x}_i^t, \xi_i^t)$ is the local gradient of client $i$ at iteration $t$, and $\mathbf{U}_i$ denotes the local part of $\mathbf{U}$ for client $i$. Clearly, in this update scheme, when $t$ is a communication iteration, we have

$$\mathbb{E}_{C_{t+1}}\left[\mathbf{X}^{t+1}\right] = \mathbb{E}_{C_{t+1}}\left[\mathbf{X}^{t+1-I} + \mathbf{U} \cdot \frac{N}{K} \sum_{i \in C_{t+1}} \left(\mathbf{y}_i^{t+1} - \mathbf{x}_i^{t+1-I}\right)\right] = \mathbf{U} \cdot \sum_{i=1}^{N} \mathbf{y}_i^{t+1} \triangleq \mathbf{Y}^{t+1}. \tag{28}$$

Additionally, $\mathbf{X}^{t+1} = \mathbf{Y}^{t+1}$ also holds when $t$ is not a communication iteration. Therefore, $\mathbf{Y}^{t+1} = \mathbb{E}[\mathbf{X}^{t+1}]$.

## C.3 Key Lemmas

**Lemma 3.**

$$\mathbb{E}\left[\frac{1}{N}\sum_{i=1}^{N}\parallel \mathbf{x}_i^t - \mathbf{X}_i^t \parallel^2\right] \leq 4\gamma^2 I^2 G^2.$$

*Proof.* FedSubAvg requires communication every $I$ iterations. Therefore, for any $t \geq 0$, there exists a $t_0 \leq t$, such that $t - t_0 \leq I - 1$ and $\mathbf{x}_i^{t_0} = \mathbf{X}_i^{t_0}$ for all $i \in N$. Then, we have

$$\begin{aligned}
&\mathbb{E}\left[\frac{1}{N}\sum_{i=1}^{N}\parallel \mathbf{x}_i^t - \mathbf{X}_i^t \parallel^2\right]\\
&=\mathbb{E}\left[\frac{1}{N}\sum_{i=1}^{N}\parallel \left(\mathbf{x}_i^t - \mathbf{X}_i^{t_0}\right) - \left(\mathbf{X}_i^t - \mathbf{X}_i^{t_0}\right) \parallel^2\right]\\
&\leq\mathbb{E}\left[\frac{1}{N}\sum_{i=1}^{N}\parallel \sum_{\tau=t_0}^{t-1}\gamma\mathbf{g}_i^\tau - \sum_{\tau=t_0}^{t-1}\gamma\mathbf{U}_i\sum_{i=1}^{N}\mathbf{g}_i^\tau \parallel^2\right]\\
&\leq 2\mathbb{E}\underbrace{\left[\frac{1}{N}\sum_{i=1}^{N}\parallel \sum_{\tau=t_0}^{t-1}\gamma\mathbf{g}_i^\tau \parallel^2\right]}_{A_1} + 2\mathbb{E}\underbrace{\left[\frac{1}{N}\sum_{i=1}^{N}\parallel \sum_{\tau=t_0}^{t-1}\gamma\mathbf{U_i}\sum_{i=1}^{N}\mathbf{g}_i^\tau \parallel^2\right]}_{A_2}.
\end{aligned}\tag{29}$$

We first focus on bounding $A_1$:

$$A_1 \leq \frac{1}{N}\sum_{i=1}^{N}\parallel \sum_{\tau=t_0}^{t-1}\gamma^2\mathbf{g}_i^\tau \parallel^2 \leq \frac{\gamma^2\left(I-1\right)^2}{N}\sum_{i=1}^{N}\parallel \mathbf{g}_i^\tau \parallel^2 \leq \gamma^2 G^2(I-1)^2.\tag{30}$$

We next bound $A_2$:

$$A_2 \leq \frac{\gamma^2(I-1)}{N}\sum_{\tau=t_0}^{t-1}\underbrace{\sum_{i=1}^{N}\parallel \mathbf{U}_i\sum_{i=1}^{N}\mathbf{g}_i^\tau \parallel^2}_{A_3},\tag{31}$$

where $A_3$ can be bounded as follows:

$$\begin{aligned}
A_3 &\leq \sum_{m=1}^{M}\sum_{m\in S(i)}\left(\frac{\sum_{m\in S(i)}\mathbf{g}_{i,\{m\}}^t}{n_m}\right)^2 = \sum_{m=1}^{M}\frac{1}{n_m}\left(\sum_{m\in S(i)}\mathbf{g}_{i,\{m\}}^t\right)^2\\
&\leq \sum_{m=1}^{M}\sum_{m\in S(i)}\left(\mathbf{g}_{i,\{m\}}^t\right)^2 \leq NG^2.
\end{aligned}\tag{32}$$

Substituting equations 30, 31, and 32 into 29 yields

$$\mathbb{E}\left[\frac{1}{N}\sum_{i=1}^{N}\parallel \mathbf{x}_i^t - \mathbf{X}^t \parallel^2\right] \leq 4\gamma^2 G^2(I-1)^2.\tag{33}$$

$\square$

## C.4 Completing the Proof of Theorem 3

*Proof.* For each client $i$, by the $L$-smoothness of $f_i(\cdot)$, we have

$$\mathbb{E}\left[f_i\left(\mathbf{X}_i^{t+1}\right)\right] \leq \mathbb{E}\left[f_i\left(\mathbf{X}_i^t\right)\right] + \underbrace{\mathbb{E}\left[\left\langle \mathbf{X}_i^{t+1} - \mathbf{X}_i^t, \nabla f_i\left(\mathbf{X}_i^t\right)\right\rangle\right]}_{C_1^i} + \frac{L}{2}\underbrace{\mathbb{E}\left[\parallel \mathbf{X}_i^{t+1} - \mathbf{X}_i^t \parallel^2\right]}_{C_2^i}.\tag{34}$$

We first focus on bounding $C_1^i$.

$$
\begin{aligned}
C_1^i =& \mathbb{E}\left[\left\langle \mathbf{Y}_i^{t+1} - \mathbf{X}_i^t, \nabla f_i\left(\mathbf{X}_i^t\right)\right\rangle\right] + \mathbb{E}\left[\left\langle \mathbf{X}_i^{t+1} - \mathbf{Y}_i^{t+1}, \nabla f_i\left(\mathbf{X}_i^t\right)\right\rangle\right] \\
=& \mathbb{E}\left[\left\langle -\gamma \mathbf{U}_i \sum_{i=1}^N \mathbf{g}_i^t, \nabla f_i\left(\mathbf{X}_i^t\right)\right\rangle\right] = \mathbb{E}\left[\left\langle -\gamma \mathbf{U} \sum_{i=1}^N \mathbf{g}_i^t, \nabla f_i\left(\mathbf{X}_i^t\right)\right\rangle\right]
\end{aligned}
\tag{35}
$$

Substituting $C_1^i$ over $i \in [N]$ yields

$$
\begin{aligned}
\mathbb{E}\left[\frac{1}{N}\sum_{i=1}^N C_1^i\right] =& -\gamma \mathbb{E}\left[\left\langle \mathbf{U}\sum_{i=1}^N \mathbf{g}_i^t, \frac{1}{N}\sum_{i=1}^N \nabla f_i\left(\mathbf{X}_i^t\right)\right\rangle\right] \\
=& -\frac{\gamma}{N}\mathbb{E}\left[\left\langle \mathbf{U}\sum_{i=1}^N \nabla f_i\left(\mathbf{x}_i^t\right), \sum_{i=1}^N \nabla f_i\left(\mathbf{X}_i^t\right)\right\rangle\right] \\
\overset{(a)}{=}& -\frac{\gamma}{N}\mathbb{E}\left[\left\langle \mathbf{V}\sum_{i=1}^N \nabla f_i\left(\mathbf{x}_i^t\right), \mathbf{V}\sum_{i=1}^N \nabla f_i\left(\mathbf{X}_i^t\right)\right\rangle\right] \\
=& -\frac{\gamma}{2N}\mathbb{E}\left[\left(\sum_{i=1}^N \nabla f_i\left(\mathbf{X}_i^t\right)\right)^\top \mathbf{U}\left(\sum_{i=1}^N \nabla f_i\left(\mathbf{X}_i^t\right)\right)\right] \\
& -\frac{\gamma}{2N}\mathbb{E}\left[\left(\sum_{i=1}^N \nabla f_i\left(\mathbf{x}_i^t\right)\right)^\top \mathbf{U}\left(\sum_{i=1}^N \nabla f_i\left(\mathbf{x}_i^t\right)\right)\right] \\
& +\frac{\gamma}{2N}\underbrace{\mathbb{E}\left[\left(\sum_{i=1}^N \left(\nabla f_i\left(\mathbf{x}_i^t\right) - \nabla f_i\left(\mathbf{X}_i^t\right)\right)\right)^\top \mathbf{U}\left(\sum_{i=1}^N \left(\nabla f_i\left(\mathbf{x}_i^t\right) - \nabla f_i\left(\mathbf{X}_i^t\right)\right)\right)\right]}_{D},
\end{aligned}
\tag{36}
$$

where $\mathbf{V} = \mathrm{diag}(1/\sqrt{n_1}, 1/\sqrt{n_2}, \cdots, 1/\sqrt{n_M})$ and $\mathbf{U} = \mathbf{V}^2$ in (a), while $D$ can be bounded as follows:

$$
D \leq \frac{1}{n_{\min}}\|\sum_{i=1}^N \left(\nabla f_i\left(\mathbf{x}_i^t\right) - \nabla f_i\left(\mathbf{X}_i^t\right)\right)\|^2 \leq \frac{NL^2}{n_{\min}}\sum_{i=1}^N \|\mathbf{x}_i^t - \mathbf{X}_i^t\|^2 \leq \frac{4N^2\gamma^2 G^2 L^2 (I-1)^2}{n_{\min}},
\tag{37}
$$

We next consider bounding $C_i^2$:

$$
C_i^2 \leq 2\mathbb{E}\underbrace{\left[\|\mathbf{Y}_i^{t+1} - \mathbf{X}_i^t\|^2\right]}_{E_1^i} + 2\mathbb{E}\underbrace{\left[\|\mathbf{X}_i^{t+1} - \mathbf{Y}_i^{t+1}\|^2\right]}_{E_2^i}.
\tag{38}
$$

Since $\mathbf{Y}_i^{t+1} = \mathbf{X}_i^t - \gamma \mathbf{U}_i \sum_{i=1}^{N} \mathbf{g}_i^t$, we have

$$
\begin{aligned}
\mathbb{E}\left[\frac{1}{N}\sum_{i=1}^{N}E_1^i\right] =& \mathbb{E}\left[\frac{1}{N}\sum_{i=1}^{N}\gamma^2 \parallel \mathbf{U}_i \sum_{i=1}^{N}\mathbf{g}_i^t \parallel^2\right] \\
=& \frac{\gamma^2}{N}\mathbb{E}\left[\left(\sum_{i=1}^{N}\mathbf{g}_i^t\right)^{\top}\mathbf{U}\left(\sum_{i=1}^{N}\mathbf{g}_i^t\right)\right] \\
=& \frac{\gamma^2}{N}\mathbb{E}\left[\left(\sum_{i=1}^{N}\left(\mathbf{g}_i^t - \nabla f_i\left(\mathbf{x}_i^t\right)\right)\right)^{\top}\mathbf{U}\left(\sum_{i=1}^{N}\left(\mathbf{g}_i^t - \nabla f_i\left(\mathbf{x}_i^t\right)\right)\right)\right] \\
& + \frac{\gamma^2}{N}\mathbb{E}\left[\left(\sum_{i=1}^{N}\nabla f_i\left(\mathbf{x}_i^t\right)\right)^{\top}\mathbf{U}\left(\sum_{i=1}^{N}\nabla f_i\left(\mathbf{x}_i^t\right)\right)\right] \\
& + \frac{2\gamma}{N}\mathbb{E}\left[\left(\sum_{i=1}^{N}\left(\mathbf{g}_i^t - \nabla f_i\left(\mathbf{x}_i^t\right)\right)\right)^{\top}\mathbf{U}\left(\sum_{i=1}^{N}\left(\mathbf{g}_i^t - \nabla f_i\left(\mathbf{x}_i^t\right)\right)\right)\right] \quad (39) \\
=& \frac{\gamma^2}{N}\mathbb{E}\left[\left(\sum_{i=1}^{N}\left(\mathbf{g}_i^t - \nabla f_i\left(\mathbf{x}_i^t\right)\right)\right)^{\top}\mathbf{U}\left(\sum_{i=1}^{N}\left(\mathbf{g}_i^t - \nabla f_i\left(\mathbf{x}_i^t\right)\right)\right)\right] \\
\leq& \frac{\gamma^2}{n_{\min}N}\mathbb{E}\left[\parallel \sum_{i=1}^{N}\left(\mathbf{g}_i^t - \nabla f_i\left(\mathbf{x}_i^t\right)\right)\parallel^2\right] \\
& + \frac{\gamma^2}{N}\mathbb{E}\left[\left(\sum_{i=1}^{N}\nabla f_i\left(\mathbf{x}_i^t\right)\right)^{\top}\mathbf{U}\left(\sum_{i=1}^{N}\nabla f_i\left(\mathbf{x}_i^t\right)\right)\right] \\
\leq& \frac{\gamma^2\sigma^2}{n_{\min}} + \frac{\gamma^2}{N}\mathbb{E}\left[\left(\sum_{i=1}^{N}\nabla f_i\left(\mathbf{x}_i^t\right)\right)^{\top}\mathbf{U}\left(\sum_{i=1}^{N}\nabla f_i\left(\mathbf{x}_i^t\right)\right)\right].
\end{aligned}
$$

If $t$ is not a communication iteration, we have $E_2^i = 0$; otherwise, we have

$$
\begin{aligned}
\mathbb{E}_{C_{t+1}}[E_2^i] =& \mathbb{E}_{C_{t+1}}\left[\parallel \mathbf{X}_i^{t_0} + \frac{N}{K}\mathbf{U}_i \sum_{j \in C_{t+1}}\left(\mathbf{y}_j^{t+1} - \mathbf{x}_j^{t_0}\right) - \mathbf{U}_i\sum_{j=1}^{N}\mathbf{y}_j^{t+1}\parallel^2\right] \\
=& \mathbb{E}_{C_{t+1}}\left[\parallel \frac{N}{K}\mathbf{U}_i \sum_{j \in C_{t+1}}\left(\mathbf{y}_i^{t+1} - \mathbf{x}_j^{t_0}\right) - \left(\mathbf{U}_i\sum_{j=1}^{N}\mathbf{y}_j^{t+1} - \mathbf{X}_i^{t_0}\right)\parallel^2\right] \\
\overset{(a)}{\leq}& \frac{N^2}{K^2}\mathbf{u}_i\mathbb{E}_{C_{t+1}}\left[\sum_{j \in C_{t+1}}\left(\mathbf{y}_i^{t+1} - \mathbf{x}_j^{t_0}\right)\circ\left(\mathbf{y}_i^{t+1} - \mathbf{x}_j^{t_0}\right)\right] \quad (40) \\
=& \frac{N}{K}\mathbf{u}_i\sum_{j=1}^{N}\left(\mathbf{y}_i^{t+1} - \mathbf{x}_j^{t_0}\right)\circ\left(\mathbf{y}_i^{t+1} - \mathbf{x}_j^{t_0}\right) \\
\leq& \frac{NI}{K}\mathbf{u}_i\sum_{j=1}^{N}\sum_{\tau=t_0}^{t+1}\gamma^2\mathbf{g}_j^{\tau}\circ\mathbf{g}_j^{\tau} \\
\leq& \frac{NI\gamma^2}{K}\mathbf{u}_i\sum_{j=1}^{N}\sum_{\tau=t-I}^{t}\mathbf{g}_j^{\tau}\circ\mathbf{g}_j^{\tau},
\end{aligned}
$$

where $\mathbf{u}_i$ denotes the local part of $(1/n_1^2, 1/n_2^2, \cdots, 1/n_M^2)$ for client $i$, $\circ$ denotes the element-wise multiplication, and (a) follows from that $\mathbb{E}[\parallel \mathbf{z} - \mathbb{E}[\mathbf{z}]\parallel^2] \leq \mathbb{E}[\parallel \mathbf{z}\parallel^2]$ holds for any random vector $\mathbf{z}$.

Summing over $i = [N]$, we have

$$
\mathbb{E}\left[\frac{1}{N}\sum_{i=1}^{N} E_2^i\right] \leq \frac{1}{N}\sum_{i=1}^{N}\frac{NI\gamma^2}{K}\mathbf{u}_i\sum_{i=j}^{N}\sum_{\tau=t-I}^{t}\mathbf{g}_j^\tau \circ \mathbf{g}_j^\tau
$$

$$
=\frac{I\gamma^2}{K}\sum_{i=1}^{N}\mathbf{u}_i\sum_{\tau=t-I}^{t}\sum_{j=1}^{N}\mathbf{g}_j^\tau \circ \mathbf{g}_j^\tau \tag{41}
$$

$$
=\frac{I\gamma^2}{K}\sum_{\tau=t-I}^{t}\sum_{i=1}^{N}\mathbf{u}_i\sum_{j=1}^{N}\mathbf{g}_j^\tau \circ \mathbf{g}_j^\tau \leq \frac{N\gamma^2 I^2 G^2}{n_{\min}K}.
$$

Taking an average of equation 34 over $i \in [N]$, substituting equations 35, 37, 38, 39, 41 into 34, and rearranging the terms yields

$$
\mathbb{E}\left[\frac{1}{N}\left(\sum_{i=1}^{N}\nabla f_i\left(\mathbf{X}_i^t\right)\right)^\top \mathbf{U}\left(\sum_{i=1}^{N}\nabla f_i\left(\mathbf{X}_i^t\right)\right)\right]
$$

$$
\leq \frac{2\left(\mathbb{E}\left[f\left(\mathbf{X}^t\right)\right] - \mathbb{E}\left[f\left(\mathbf{X}^{t+1}\right)\right]\right)}{\gamma} + \frac{2\gamma L\sigma^2}{n_{\min}} + \frac{4N\gamma^2 G^2 L^2(I-1)^2}{n_{\min}} + \frac{2\gamma NI^2 G^2 L}{n_{\min}K}. \tag{42}
$$

Summing over $t \in \{1, 2, \cdots, T\}$ and dividing both sides by $T$ yields

$$
\mathbb{E}\left[\frac{1}{T}\sum_{i=1}^{T}\frac{1}{N}\left(\sum_{i=1}^{N}\nabla f_i(\mathbf{X}^t)\right)^\top \mathbf{U}\left(\sum_{i=1}^{N}\nabla f_i(\mathbf{X}^t)\right)\right]
$$

$$
\leq \frac{2\left(f\left(\mathbf{X}^1\right) - f\left(\mathbf{X}^*\right)\right)}{\gamma T} + \frac{2\gamma L\sigma^2}{n_{\min}} + \frac{4N\gamma^2 I^2 G^2 L^2}{n_{\min}} + \frac{2\gamma NI^2 G^2 L}{n_{\min}K}. \tag{43}
$$

Therefore, we have

$$
\mathbb{E}\left[\frac{1}{T}\sum_{i=1}^{T}\nabla f(\mathbf{X}^t)^\top \mathbf{D}\nabla f(\mathbf{X}^t)\right]
$$

$$
=\mathbb{E}\left[\frac{1}{T}\sum_{i=1}^{T}\frac{1}{N}\left(\sum_{i=1}^{N}\nabla f_i(\mathbf{X}^t)\right)^\top \mathbf{U}\left(\sum_{i=1}^{N}\nabla f_i(\mathbf{X}^t)\right)\right] \tag{44}
$$

$$
\leq \frac{2\left(f\left(\mathbf{X}^1\right) - f\left(\mathbf{X}^*\right)\right)}{\gamma T} + \frac{2\gamma L\sigma^2}{n_{\min}} + \frac{4N\gamma^2 I^2 G^2 L^2}{n_{\min}} + \frac{2\gamma NI^2 G^2 L}{n_{\min}K}.
$$

$\square$

# D    Experimental Details

## D.1    Feature Heat Distributions on Datasets

Figure 5 shows the feature heat distributions of the head features on four datasets in NLP or RS. We can observe that the feature heat (i.e., the number of feature-involved clients) varies widely among different features.

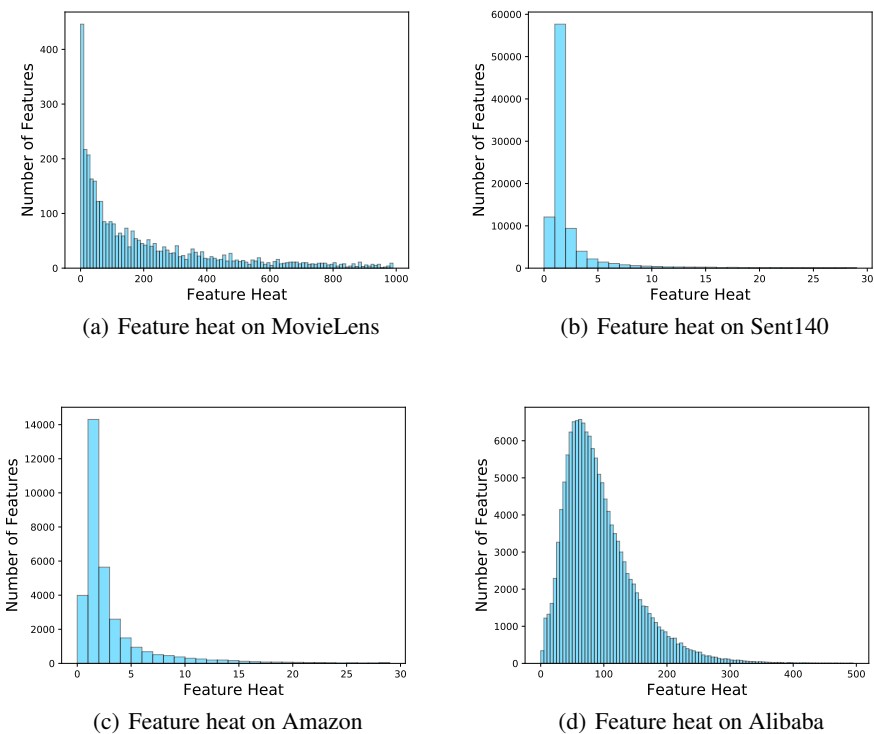

(a) Feature heat on MovieLens

(b) Feature heat on Sent140

(c) Feature heat on Amazon

(d) Feature heat on Alibaba

Figure 5: Feature heat distributions on four datasets (only the head features are shown). The x-axis represents feature heat, i.e., the number of clients involving a movie/word/item, and the y-axis represents the number of movies/words/items under a certain heat.

## D.2    The Approximation of Scaffold in CTR Prediction

In CTR prediction, we make an approximation to Scaffold since a resource-constrained client cannot keep the control variate. Therefore, we perform the controlled update on the cloud server in each round. Let $c$ denote the globally controlled variate, and let $c_i$ denote client $i$'s local control variate. At the end of each round in Scaffold, we have:

$$c^{old} \approx \frac{1}{N} \sum_{i=1}^{N} c_i^{old}, \quad c^{new} \leftarrow c^{old} + \frac{1}{N} \sum_{i \in S_c} (c_i^{new} - c_i^{old}), \tag{45}$$

where $S_c$ is the selected clients at the round. Taking expectation with respect to the selected clients, we have

$$\mathbb{E}[c^{new}] = c^{old} + \frac{|S_c|}{N^2} \sum_{i=1}^{N} (c_i^{new} - c_i^{old}) \approx \frac{N - |S_c|}{N} c^{old} + \frac{|S_c|}{N} \cdot \frac{1}{N} \sum_{i=1}^{N} c_i^{new}$$

$$\approx \frac{N - |S_c|}{N} c^{old} + \frac{|S_c|}{N} \cdot \mathbb{E}\left[ \frac{1}{|S_c|} \sum_{i \in S_c} c_i^{new} \right]. \tag{46}$$

In addition, the global update $\Delta \mathbf{X} \approx -\eta Ic$ and client $i$'s local update $\Delta \mathbf{x}_i \approx -\eta Ic_i$. By equation 46, we can approximate the global update by

$$\Delta \mathbf{X}^{new} \approx \frac{N - |S_c|}{N} \Delta \mathbf{X}^{old} + \frac{|S_c|}{N} \left( \frac{1}{|S_c|} \sum_{i \in S_c} \Delta \mathbf{x}_i \right). \tag{47}$$

Therefore, we run Scaffold approximately by weighted averaging the original global update and the aggregated local updates to get the new global update every communication round.

## D.3 Hyperparameters

For the tasks of rating classification and sentiment analysis, we set the local batch size to 5 and set the local iteration number to 10 in all FL algorithms. We set the batch size to 250 and set the iteration number in each round to 10 in CentralSGD. For the CTR prediction on Amazon, we set the batch size to 4 and set the local iteration number to 10 in all FL algorithms. We set the batch size to 400 and set the iteration number in each round to 10 in CentralSGD. For the CTR prediction on the Alibaba dataset, we set the batch size to 32 and set the local iteration number to 10 in all FL algorithms. We set the batch size to 3,200 and set the iteration number in each round to 10 in CentralSGD. We search the learning rate for each algorithm independently, and the learning rates are recorded in Table 4. In addition, we tune the hyperparameters for FedAdam and list the hyperparameters in Table 5.

Table 4: Learning rate for each experiment.

|  | CentralSGD | FedAvg | FedProx | Scaffold | FedSubAvg |
|---|---|---|---|---|---|
| MovieLens | 0.1 | 0.1 | 0.1 | 0.1 | 0.1 |
| Sent140 | 0.1 | 0.1 | 0.1 | 0.1 | 0.1 |
| Amazon | 0.05 | 0.1 | 0.1 | 0.1 | 0.05 |
| Alibaba | 1 | 1 | 1 | 1 | 0.3 |

Table 5: Hyperparameters for FedAdam.

|  | $\eta_l$ | $\eta$ | $\beta_1$ | $\beta_2$ |
|---|---|---|---|---|
| MovieLens | 0.1 | 1 | 0.9 | 99 |
| Sent140 | 1 | 1 | 0.9 | 0.99 |
| Amazon | 0.1 | 0.001 | 0.9 | 0.999 |
| Alibaba | 1 | 0.001 | 0.9 | 0.99 |

## D.4 Supplementary Notes for the Experiments

In our experiments, all FL algorithms are extended to the weighted case. In particular, the correction coefficient $N/n_m$ for parameter $m$ in FedSubAvg is extended to $\sum_{i=1}^{N} w_i / \sum_{\{j|m \in S(j)\}} w_j$, where $w_i$ is the size of client $i$'s local training data. For the rating classification, the MovieLens dataset is available from https://grouplens.org/datasets/movielens/1m/. We randomly select 20% of the samples as the test dataset and leave the remaining 80% as the training set, and further randomly choose 10,000 samples from the training set to evaluate train losses. For the sentiment analysis, the Sentiment140 dataset is available from http://help.sentiment140.com/for-students, and we randomly select 20% of the samples as the test dataset and leave the remaining 80% as the training set. For the CTR prediction, the Amazon dataset is available from http://jmcauley.ucsd.edu/data/amazon/, and we partition the dataset based on the timestamp. In addition, experiments are conducted on machines with operating system Ubuntu 18.04.3 and one NVIDIA GeForce RTX 2080Ti GPU.

## D.5 Additional Results

We show the test accuracies (ACCs) or AUCs for each experiment. Figure 6 compares the test ACCs or AUCs of FedSubAvg and baselines under default settings. Figure 7 compares the test ACCs or

AUCs of FedSubAvg with different numbers of participating client per round $K$. All the results from test ACC or AUC are consistent with the results from the train loss.

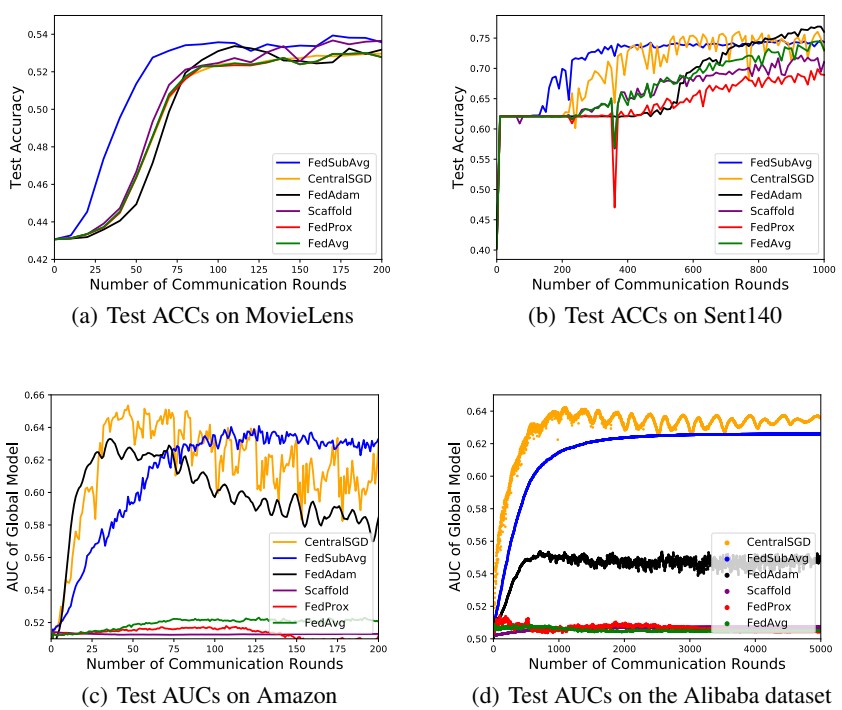

(a) Test ACCs on MovieLens

(b) Test ACCs on Sent140

(c) Test AUCs on Amazon

(d) Test AUCs on the Alibaba dataset

Figure 6: Test ACCs or test AUCs of FedSubAvg and the baselines on different datasets.

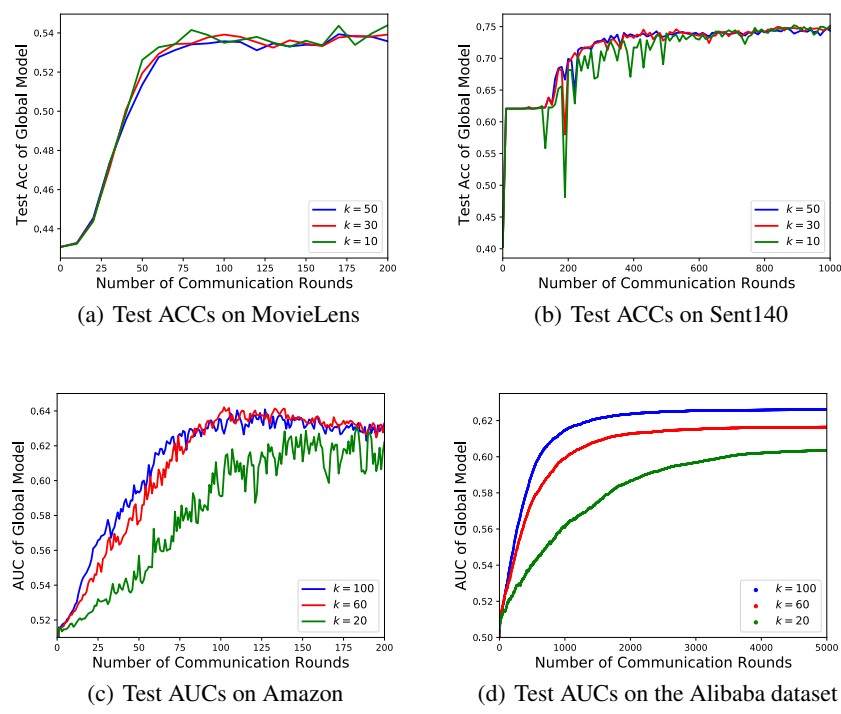

(a) Test ACCs on MovieLens

(b) Test ACCs on Sent140

(c) Test AUCs on Amazon

(d) Test AUCs on the Alibaba dataset

Figure 7: Test ACCs or test AUCs of FedSubAvg on different datasets with the varying number of selected clients per round $K$.

## E  Comparison with Adaptive Federated Optimization

In this section, we compare FedSubAvg with FedAdam [26] and show that in the federated settings with feature heat dispersion, FedSubAvg generally has stronger theoretical guarantees and lower computation overhead.

We first clarify the relationship of FedSubAvg and the adaptive federated algorithms from the perspective of algorithm design. FedSubAvg is a prior preconditioning method to handle the issue of feature heat dispersion, which is newly identified and proven to cause ill-condition problem. The diagonal preconditioner in FedSubAvg is to do re-weighting based only on the statistics over the clients' local data and keeps unchanged during federated learning. In contrast and in parallel, FedAdam uses posterior information (i.e., the estimates of the first and second moments of the global updates), which is dynamic in the federated optimization process, to apply Adam in FL settings, and may alleviate the issue of feature heat dispersion.

We next compare FedSubAvg with FedAdam from the perspective of theoretical analysis. In this paper, we theoretically prove that FedSubAvg works as a suitable preconditioner for the ill-conditioned global objective caused by feature heat dispersion. In contrast, FedAdam has no strict theoretical guarantees in terms of reducing the condition number. In addition, FedAdam has a $O(1/\sqrt{NT})$ convergence rate with respect to $\| \nabla f(\mathbf{X}) \|^2$ **when assuming full participation** (i.e., $K = N$). By Theorem 5, FedAdam only has a convergence guarantee of $O(\sqrt{\frac{N}{n_{\min}^2 T}})$ with respect to $f(\mathbf{X}_{local}^*)$, which indicates that compared to FedAdam, FedSubAvg converges $\Theta(\sqrt{N/n_{\min}})$ times faster.

We further compare FedSubAvg with FedAdam from the perspective of computation overhead. Since the prior precondition in FedSubAvg keeps unchanged during training, the additional computation complexity is $O(M)$, where $M$ is the number of model parameters. In contrast, FedAdam calculates the adaptive learning rates for each model parameter every communication round, which leads to additional $O(RM)$ computation complexity, where $R$ is the number of rounds. Therefore, the additional computational overhead of FedSubAvg is significantly smaller than that of FedAdam, especially when $M$ and $R$ are large.

## F  Privacy Preserving Methods

Regarding the privacy issues in federated submodel learning, Niu et al. [20] designed a protocol based on private set union, randomized response [27], and secure aggregation [28], which can protect each individual client's local features (i.e., the position of its submodel in the full model) with strict local differential privacy guarantee in both download and upload phases, against the cloud server and any other client. Compared with [20], the additional information needed in FedSubAvg is how many clients have each individual feature, thereby computing the feature heat dispersion and the diagonal pre-conditioner. To obtain such information without revealing any client's local features, one feasible way is to apply secure aggregation, where each client uses a vector to truly indicate whether it has feature $i$ in the $i$-th position of the vector, and the cloud server can accurately obtain the sum of all the clients' vectors without any individual client's vector and further can obtain the size of clients having each individual feature. Another more efficient and an unbiased way is to apply randomized response, where each client still uses a vector to indicate whether it has feature $i$ in the $i$-th position of the vector, but the difference is that, conditional only whether the client truly has feature $i$, it will indicate "1" with a certain probability and "0" with another probability. Based on the randomized vectors from all the clients, the cloud server can obtain an unbiased estimation of how many clients having each individual feature after certain corrections. Meanwhile, each client can hold plausible deniability (in terms of local differential privacy) against whether it has a certain feature.