# OpenReview forum: "Federated Submodel Optimization for Hot and Cold Data Features"
_NeurIPS.cc/2022/Conference — NeurIPS 2022 Accept_

### Official Review · Reviewer_9rr3 · 2022-07-08

**Rating:** 6
**Confidence:** 4
**Soundness:** 3 good
**Presentation:** 3 good
**Contribution:** 3 good

**Summary:**

The authors point out the fact that (in the context of federated learning) client’s local data normally involve a small subspace of the full feature space. Especially in the case of models that contain large sparse embeddings, this would mean that each client downloads and updates only a small part of the full global model (I.e., a submodel).

As some features are more popular than others (e.g., words in a vocabulary), some embedding will then be averaged by a larger fraction of clients than others. The authors then show that this discrepancy (called heat dispersion) might result in slower convergence of algorithms like FedAvg.

They then propose a new method where each weight essentially has a different learning rate, based on how many clients participate in its update. They show both analytically and through an evaluation that this method improves the convergence speed.




**Questions:**

Overall this is a great method to address sparse feature dispersion in FL.

- Practically this algorithm works by providing a different averaging coefficient for each parameter, based on the "popularity" of each feature amongst the clients. As a result, the resulting overall (average) learning rate might be affected.  While this shows faster convergence on the empirical evaluation, it is unclear if this is due to using a default learning rate (and the same learning rate) across each method. A more fair experiment would be to make sure that a hyper-parameter search is conducted for each baseline and the proposed method. I am mentioning this because practically, by changing the averaging coefficient, FedAvg might be more tolerant to higher LR (thus improving convergence speed) than FedSubAvg before convergence is affected. It would be great if the authors revised their experiments and show that FedSubAvg is able to beat the other methods when each method is using hyperparams that are searched independently.

- The second question is wrt to the number of clients (K) vs other methods. The authors show that as the number of clients increase, FedSubAvg converges faster. As in many real-world FL deployed systems K = thousands of devices, it would be great to be able to also include the results from the baselines (or at least fedAvg) and possibly show for values > 100. Its unclear if the convergence of FedSubAvg is much better that the SOTA only for small K or if this algorithm would actually improve convergence in real-world scenarios.

- The third question is about combining this method with privacy-preserving mechanisms. There were many approaches where secure embedding retrieval's were used, and local DP noise is added to the model to make sure that user privacy and regulations are respected. It would be great if the authors comment on the compatibility of their approach with such privacy-preserving approaches, especially as they have to be able to identify the exact features that the clients used to update the model.

**Limitations:**

Please see above (questions)

**Strengths And Weaknesses:**

+ The authors address an interesting problem on training FL models with sparse embeddings: the fact that not all of them are equally popular.

+ The authors conducted a theoretical analysis of their model

+ They compared with 4 baselines, some of them designed to speed up convergence on non-iid data.

+ The results show that their methodology is promising

- There are some assumptions in the evaluation (see below in questions for details)

- It is unclear how well this methodology would work with privacy-preserving mechanisms (e.g., local DP noise)

---

> ### Author Response · Authors · 2022-07-30
> **Response to Reviewer 9rr3**
>
> We thank the reviewer for appreciating our key contributions. We hope the following response can fully address your comments.
>
> First, we clarify that we have searched learning rates for FeAvg and FedSubAvg independently, including trying lr=10 for FedAvg on different datasets. For Amazon and Alibaba datasets, the chosen learning rate for FedAvg is larger than that of FedSubAvg (cf. Table 4).
>
> Second, we clarify that $K$ denotes the number of chosen clients per round rather than the total number of clients $N$. In our experiments, $N = 49,023$ for the Alibaba dataset, and $N > 1000$ for other datasets. In Google's real-world FL system on Gboard [1], $K = 100$, which is included in our experiments.
>
> Third, we clarify that FedSubAvg is compatible with privacy preserving approaches. In particular, for the privacy issues in federated submodel learning, [2] have designed a protocol based on private set union, randomized response [3], and secure aggregation [4], which can protect each individual client’s local features (i.e., the position of its submodel in the full model) with strict local differential privacy guarantee in both download and upload phases, against the cloud server and any other client. Compared with [2], the additional information needed in the proposed FedSubAvg algorithm is how many clients have each individual feature, thereby computing the feature heat dispersion and the diagonal pre-conditioner. To obtain such information without revealing any client’s local features, one feasible way is to apply secure aggregation [4], where each client uses a vector to truly indicate whether it needs (i.e., its submodel contains) parameter $i$ in the $i$-th positon of the vector, and the cloud server can accurately obtain the sum of all the clients’ vectors without any individual client’s vector and further can obtain the size of clients needing each individual parameter. Another more efficient way is to apply randomized response [3], where each client still uses a vector to indicate whether it needs parameter $i$ in the $i$-th positon of the vector, but the difference is that, based on whether the client truly needs parameter $i$, it will indicate “1” with a certain probability and “0” with another probability. Collecting the randomized vectors from all the clients, the cloud server can obtain an unbiased estimation of how many clients need each individual parameter after certain corrections. Meanwhile, each client can hold plausible deniability (in terms of local differential privacy) against whether it needs a certain parameter. We also note that the unbiased estimation of diagonal pre-conditioner, which is independent from the training process, will not affect the convergence analysis of FedSubAvg.
>
> [1] Towards Federated Learning at Scale: System Design, in MLSys, 2019.
>
> [2] Billion-Scale Federated Learning on Mobile Clients: A Submodel Design with Tunable Privacy, in MobiCom, 2020.
>
> [3] Randomized Response: A Survey Technique for Eliminating Evasive Answer Bias. In Journal of the American Statistical Association, 1965.
>
> [4] Practical Secure Aggregation for Privacy-Preserving Machine Learning, in CCS, 2017.

---

### Official Review · Reviewer_Me5H · 2022-07-11

**Rating:** 5
**Confidence:** 4
**Soundness:** 2 fair
**Presentation:** 2 fair
**Contribution:** 3 good

**Summary:**

The authors consider a specific federated learning scenario where different data features ‘involve’ different clients. For some features, a large number of clients can be involved while other features might involve only limited clients. The authors show that in this case, the classical FedAvg can suffer from slow convergence. In the proposed new algorithm, the aggregations of parameter updates are weighted per parameter by the ratios of the local clients involved. The authors prove that by reweighting the parameter updates in this way, the condition numbers of the Hessian of the learning objectives become smaller than the original Hessian. In the experiments with four real-world datasets, the authors demonstrated that the proposed algorithm (FedSubAvg) offers faster convergence than existing alternatives.


**Questions:**

Please see Strengths And Weaknesses section.

**Limitations:**

The limitations and potential negative societal impact were not discussed.

**Strengths And Weaknesses:**

Strengths:
Improving the convergence of federated learning is an important and active area of research. The authors contribute a new algorithm that can potentially improve the convergence of the federated averaging algorithm. The experiments were conducted on large-scale real-world datasets.

Limitations:
- The paper would need a thorough rewriting. For example, the example illustration of feature heat dispersion in recommender systems at L45-50 is difficult to comprehend. What does ‘less than 1% of the average’ mean? Please formally define ‘involvement‘. L122 ‘the number of clients who involve this model parameter’: How is this number determined in general?
- The proposed algorithm seems applicable only when ‘submodels‘ are well-defined, i.e., the individual clients do not have to update the full model parameters but instead, can download and update only the required small parts (submodel) of the complete model. On the other hand, typical local learning steps tend to require simultaneous updates of all model parameters, unless the model is linear. The authors should provide a detailed discussion as to when such submodels are well-defined and how the index set S(i) is determined in practice. Can this algorithm be applied to the standard MLPs?

Minor comments
- The proposed algorithm can be considered as diagonal preconditioning on stochastic gradient descent (SGD) (L167). The authors could discuss connections to existing SGD preconditioning methods, e.g., AdaGrad: AdaGrad preconditions the SGD update based on the magnitudes of individual parameter updates. Extending AdaGrad to the federated learning setting can be straightforward.
- Please enlarge the plots in figures 3 and 4.

---

> ### Author Response · Authors · 2022-07-30
> **Response to Reviewer Me5H**
>
> We address your comments as follows.
>
> First, we restate the example in recommender systems. For cold item 1 only appearing in 1% of clients' (denoted as client group A) local datasets, the corresponding embedding vector for item 1 is only contained by the submodels of those clients in group A. In conventional FedAvg, the update of the embedding vector for item 1 will be significantly slowed down, since only the clients in group A will upload non-zero updates. In contrast, for another hot item 2 appearing in all the clients' local datasets, the update of the corresponding embedding vector will not be slowed down. This phenomenon leads to different parameters of the global model being optimized at different speeds.
>
> Second, we emphasize that we focus on the recommendation and NLP tasks in this work (line 73), where the deep models typically contain a large embedding layer and some other dense layers (e.g., standard MLPs as you mentioned). **The main bottleneck of supporting these deep models in cross-device FL is the size of the the embedding layer rather than the size of the dense layers.** For example, the embedding layer of the popular deep interest network (DIN) [1] for recommendation tasks is larger than 100GB, far beyond any mobile device's capacity, while the dense layers are smaller than 100KB, easy to be deployed on mobile devices. Therefore, the submodel selection method (lines 81 - 86), which **retrieves a few embedding vectors for the client's local item ids in a key-value lookup way and directly takes the other dense layers**, is an effective and efficient partition technique for the large embedding layer and makes cross-device FL possible. In practice, a client can determine the index set of its submodel (i.e., the parameters in the retrieved embeddings and the other full dense layers) based on its local dataset before FL begins. For example, in recommendation (resp., NLP) scenarios, the client's local item ids (resp., word ids) and the indexes of other dense layers function as the index set of its submodel.
>
> Third, we clarify that we have compared FedSubAvg with an adaptive optimizer. In particular, the work [2] has extended both AdaGrad and Adam to the FL setting, where their performances are similar. We have compared FedSubAvg with FedAdam in Appendix E of the supplementary material. We have revealed that FedAdam cannot address the issue of hot vs. cold features in FL, and meanwhile, FedSubAvg has much better model performance and much lower computation overhead than FedAdam.
>
> [1] Deep interest network for click-through rate prediction, in KDD, 2018.
>
> [2] Adaptive federated optimization, in ICLR, 2021.

---

### Official Review · Reviewer_wqvK · 2022-07-11

**Rating:** 5
**Confidence:** 3
**Soundness:** 2 fair
**Presentation:** 2 fair
**Contribution:** 2 fair

**Summary:**

The manuscript considers a specific use case of federated learning in a recommender system or NLP scenario and proposes to scale the model update per coordinate through the ratio between the total number of clients and the number of clients who involve this model parameter coordinate. Some theoretical results are provided to motivate and justify the necessity of design choices.

**Questions:**

Check Weaknesses part.

**Limitations:**

Check Weaknesses part.

**Strengths And Weaknesses:**

# Strengths
* The considered scenario is interesting and important to the community. The proposed method is simple yet (intuitively) effective in alleviating the identified limitation.
* The entire manuscript is generally well-structured, and most claims are well-supported.
* Extensive numerical results are provided for some aspects.

# Weaknesses
1. The index set $S(i)$ determines the sub-model of $X S(i)r$. However, for the four considered datasets, the exact distribution of features (hot vs. cold features) is still unknown, and some specific treatments (line 211- line 212, line 226 - line 227) may require some justifications, such as "we label the samples with the ratings of 4 and 5 to be positive and label the rest to be negative". Will these treatments magnify the large imbalance phenomenon between hot and cold features?
2. Given the considered challenges and proposed coordinate-wise scaling solution, one strong competitor should be considered, i.e., using an adaptive optimizer on the server as in [1].
3. The client's index set must be sent to the server, and a clustering-based FL technique can utilize this information naturally. The reviewer is also interested in how cluster-based FL approaches perform, as these algorithms may have already addressed the issue of hot vs. cold features.

# Reference
1. Adaptive Federated Optimization

---

# Post rebuttal
The reviewer acknowledged the authors' feedback and checked other reviews.

The response has addressed concerns #2 and (partially) #3. However, the reviewer still believes that the manuscript needs some revisions to polish its text (e.g., make it self-contained, well-structured, precise, etc).

---

> ### Author Response · Authors · 2022-07-30
> **Response to Reviewer wqvK**
>
> We address your comments as follows.
>
> First, we clarify that the exact distribution of features is determined once the dataset is partitioned among clients. The treatments (lines 211-212, 226-227) are the common processing on the labels [1], which do not affect the features, thus not magnifying the imbalance phenomenon between hot and cold features.
>
> Second, we clarify that we have compared the proposed FedSubAvg with FedAdam in the previous work as you mentioned [2] in Appendix E of the supplementary material. We have revealed that FedAdam cannot address the issue of hot vs. cold features in FL, and meanwhile, FedSubAvg has much better model performance and much lower computation overhead than FedAdam.
>
> Third, we clarify that clustering cannot addressed the issue of hot vs. cold features. In particular, a client's dataset can have both hot and cold features. Therefore, clustering-based FL at the level of clients cannot eliminate the feature dispersion.
>
> [1] Deep interest network for click-through rate prediction, in KDD, 2018.
>
> [2] Adaptive federated optimization, in ICLR, 2021.

---

> ### Author Response · Authors · 2022-08-03
> **Response to Your Comments in Post rebuttal**
>
> Thank you very much for reading our response in detail and kindly, open-mindly raising your rating! We are quite glad that our response can fully address your concern #2. We still want to take this precious chance to clarify your remaining concerns, potentially motivating you to raise your rating further.
>
> For your concern #1, we clarify that the dataset pre-processing in our evaluation is over the **labels** **rather than** over the **features**, thus **not affecting** the imbalance phenomenon between hot and cold features as you concerned.
>
> For your concern #3, we clarify that, in essence, the **feature heat dispersion in FL** can be globally viewed from a **brand new** perspective of **"features interacts with different clients"**, whereas **existing FL work for non-iid data** (e.g., clustering-based FL as you commented) can be globally viewed from **"clients interacts with different features"**. Therefore, clustering-based FL or other existing FL work for non-iid issue **cannot** address the feature heat dispersion considered in this work. Another intuitive explaination is that **one client can have both hot and cold features** (e.g., in recommender systems, a user not only clicks some hot goods but also clicks some cold goods; in NLP scenarios, a user not only types some hot words but also types some cold words, etc.). This implies that clustering-based FL  at the level of clients can mitigate/exploit the disperation among clients, but cannot address the feature heat disperation.
>
> One potential issue that comes along with concern #3 is that "the client's index set must be sent to the server", which may rasise privacy issue. We clarify (although our focus in this work is federated submodel optimization rather than privacy preservation) that the feature heat disperation can be easily obtained **without revealing each client's real index set** with secure aggregation or randomized response. The details are as follows:
>
> "We clarify that the cloud server can obtain the feature heat dispersion without revealing any client’s real index set. One feasible way is to apply secure aggregation [1], where each client uses a vector to truly indicate whether it needs (i.e., its submodel contains) parameter  in the $i$-th position of the vector, and the cloud server can accurately obtain the sum of all the clients’ vectors without any individual client’s vector and further can obtain the number of clients needing each individual parameter. Another more efficient way is to apply randomized response [2], where each client still uses a vector to indicate whether it needs parameter  in the $i$-th position of the vector, but the difference is that, based on whether the client truly needs parameter , it will indicate “1” with a certain probability and “0” with another probability. Collecting the randomized vectors from all the clients, the cloud server can obtain an unbiased estimation of how many clients need each individual parameter after certain corrections. Meanwhile, each client can hold plausible deniability (in terms of local differential privacy) against whether it needs a certain parameter. We also note that the unbiased estimation of diagonal pre-conditioner, which is independent from the training process, will not affect the convergence analysis of FedSubAvg."
>
> [1] Practical Secure Aggregation for Privacy-Preserving Machine Learning, in CCS, 2017.
>
> [2] Randomized Response: A Survey Technique for Eliminating Evasive Answer Bias. In Journal of the American Statistical Association, 1965."
>
> If you have other comments or concerns, please feel free to raise.

---

### Official Review · Reviewer_XsFL · 2022-07-15

**Rating:** 4
**Confidence:** 3
**Soundness:** 2 fair
**Presentation:** 2 fair
**Contribution:** 2 fair

**Summary:**

This paper proposes a submodel optimization technique in federated learning in the presence of diverse data feature distributions across clients. Heterogenity in features of local data leads to distinct update speed for individual feature-related parameters, degenerating convergence of the model. The paper claims that the global objective function of typical federated learning framework is ill-conditioned when each client updates and communicate only subset of the total model (submodel).
This paper handles the problem by compensating the amount of parameter updates based on the feature heat dispersion. It also provides formal analysis for the convergence of the proposed method. It demonstrates the effectiveness of the proposed method over baselines on three benchmarks; rating classification, text sentiment classification, click-through rate prediction.



**Questions:**

- Comparison with the typical FedAvg (participating clients update all parameters of server model then aggregate them) in terms of total communicated parameters would validate the efficiency of the proposed method.



**Limitations:**

- As mentioned above, the proposed algorithm is incompatible with modern deep learning architectures (CNN, MLP), the more compatible approach should be proposed without using prior knowledge that the feature dispersion is known and feature-related parameters are predefined.

- The writing is hard to follow, and some details (e.g. implementation details) were not explained well.


**Strengths And Weaknesses:**


### Strengths

- The submodel optimization setup where each client updates the subset parameters of the central server is realistic and would be the proper extension with clear motivation. In real-world applications edge devices are likely to have limited capacity in memory, computation, and communication bandwidth.

- Experimental results are quite compelling. Figure 3 presents that FedSubAvg converges fast and shows similar tendency with centralized training (CentralSGD)

### Weakness

- The assumption that server knows indexes for the feature-related parameters of all clients is unrealistic.
Due to the assumption, the proposed algorithm is not compatible with deep models (e.g. DNN, MLP) with multiple non-linear layers.

- In the similar vein, The proposed method needs to calculate data feature dispersion, which can bring about privacy concerns.

---

> ### Author Response · Authors · 2022-07-30
> **Response to Reviewer XsFL**
>
> We address your comments as follows.
>
> First, we emphasize that we focus on the recommendation and NLP tasks in this work (line 73), where the deep models typically contain a large embedding layer and some other dense layers (e.g., MLP). **The main bottleneck of supporting these deep models in cross-device FL is the size of the the embedding layer rather than the size of the dense layers.** For example, the embedding layer of the popular deep interest network (DIN) [1] for recommendation tasks is larger than 100GB, far beyond any mobile device's capacity, while the dense layers are smaller than 100KB, easy to be deployed on mobile devices. Therefore, the submodel selection method (lines 81 - 86), which **retrieves a few embedding vectors for the client's local item ids in a key-value lookup way and directly takes the other dense layers**, is an effective and efficient partition technique for the large embedding layer and makes cross-device FL possible.
>
> Second, we clarify how each client can obtain its feature-related parameters. As stated in lines 81 - 86, a client can determine its indexes of feature-related parameters (i.e., the parameters in the retrieved embeddings and the other full dense layers) based on its local dataset before FL begins. For example, in recommendation (resp., NLP) scenarios, the client's local item ids (resp., word ids) and the indexes of other dense layers function as the index set of its submodel.
>
> Third, we clarify how the cloud server can obtain the feature heat dispersion without revealing any client’s real index set. One feasible way is to apply secure aggregation [2], where each client uses a vector to truly indicate whether it needs (i.e., its submodel contains) parameter $i$ in the $i$-th position of the vector, and the cloud server can accurately obtain the sum of all the clients’ vectors without any individual client’s vector and further can obtain the number of clients needing each individual parameter. Another more efficient way is to apply randomized response [3], where each client still uses a vector to indicate whether it needs parameter $i$ in the $i$-th position of the vector, but the difference is that, based on whether the client truly needs parameter $i$, it will indicate “1” with a certain probability and “0” with another probability. Collecting the randomized vectors from all the clients, the cloud server can obtain an unbiased estimation of how many clients need each individual parameter after certain corrections. Meanwhile, each client can hold plausible deniability (in terms of local differential privacy) against whether it needs a certain parameter. We also note that the unbiased estimation of diagonal pre-conditioner, which is independent from the training process, will not affect the convergence analysis of FedSubAvg.
>
> Fourth, we clarify that FedSubAvg indeed sharply reduces communication overhead compared with FedAvg. In particular, in industrial recommendation scenarios, each client in FedAvg uses the full model, the size of which is larger than 100GB, whereas each client in FedSubAvg uses a submodel, the size of which is normally less than 1MB.
>
> [1] Deep interest network for click-through rate prediction, in KDD, 2018.
>
> [2] Practical Secure Aggregation for Privacy-Preserving Machine Learning, in CCS, 2017.
>
> [3] Randomized Response: A Survey Technique for Eliminating Evasive Answer Bias. In Journal of the American Statistical Association, 1965.

---

> > ### Comment · Reviewer_XsFL · 2022-08-09
> > **Thanks for the author's response**
> >
> > I appreciate the authors' response to my concerns. While some of my concerns have been addressed, I still have a concern that the proposed method is not a general optimization method for federated learning but is limited to NLP and recommendation tasks where each client can identify feature-related parameters. Also, the presentation should be improved a lot. So, I will keep my score as before.

---

> > > ### Author Response · Authors · 2022-08-10
> > > **Response to Your Concerns in Post Rebuttal**
> > >
> > > Thank you for reading our response near the end of the rebuttal phase. We believe that we have addressed all your comments quite well. We stress some key points as follows.
> > >
> > > First, we stress that NLP and recommendation have been recognized as **quite important fields** of deep learning **in both academia and industry**, which should be a common sense. In theses two fields, we identified **a brand new issue of feature heat dispersion**. Our proposed federated submodel optimization method, **for the first time**, provides theoretical and technical guarantees for the application of FL in these two fields, from the perspective of **mitigating feature heat dispersion**. We believe that it is clearly a **quite important contribution**. We also stress that **MLP and other dense networks as you commented do not suffer from feature heat dispersion and are easy to be deployed on the mobile devices, thus not a practical concern of FL.**
> > >
> > > Second, we stress that we have uploaded the revised version to OpenReview (please see our post at the top). In particular, we have proof-read our manuscript for several times and improved the presentation, especially the introduction and the problem formulation sections, where the tasks, the model/submodel structure, and the feature heat dispersion have been clearly introduced. **If you have any detailed comment on the presentation, please feel free to raise** and we will certainly embody in the camera-ready version.

---

### Author Response · Authors · 2022-08-09
**Note At the End of the Rebuttal Phase**

Dear Reviewers,

We believe that we have addressed all the concerns raised and also have uploaded a revised version (the full paper is in the supplementary material).

Thank you very much for your time and effort devoted to reviewing our work! We truly hope that our work can receive a fair and convincing decision.

Best regards,

Paper5390 Authors

---

### Meta-Review · Area_Chair_Z5Cd · 2022-08-23

**Recommendation:** Accept
**Confidence:** Certain

**Metareview:**

This paper considers a particular FL scenario, where the model includes a large embedding layer as is typical in NLP and recommendation models. To make training feasible or more efficient, the FedSubAvg method is proposed. In particular, it deals with the setting where not all features are equally encountered in training data. This is leveraged to reduce communication and computation overhead, and also to improve optimization dynamics. The proposed approach comes with theoretical guarantees, and the paper also provides a thorough numerical evaluation demonstrating benefits over other approaches.

The reviewers raised concerns about the relevance and potential narrowness of the setup, assumptions, and whether the proposed FedSubAvg approach would be comparable with privacy-enhancing technologies like secure aggregation and differential privacy. It is clear that the setup considered is indeed relevant given the prevalence of models with large embedding layers in NLP and recommendation models, and the useful of such models in several applications. Given this, it isn't necessary for the authors to demonstrate any relevance to training standard MLPs since that isn't the focus and no claims are made in the paper about such architectures. The authors responses also were convincing that the approach can be made compatible with DP and secure aggregation in a reasonable way.

I'm happy to recommend that this paper be accepted. When preparing the camera ready, to make the paper accessible to a broader audience, it would be helpful to include (in the intro, or early in the paper) additional material and references to motivate the relevance of models with large embedding layers, in addition to the key revisions already made in response to the initial reviews.

**Award:**

No

---

### Decision · Program_Chairs · 2022-09-14

Accept